# Fabrication of cytotoxic mirror image nanopores

Neilah Firzan CA[1,2,7], Kalyanashis Jana [3,7], Sreelakshmi Radhakrishnan[4,5,7], Rifat Aara [6,7], Mubeena S[6], Radhika Nair [6], Harsha Bajaj [4,5], Ulrich Kleinekathöfer [3] & Kozhinjampara R. Mahendran [1] ✉

Synthetic nanopores composed of mirror-image peptides have been reported, but not fully functional mirror-image pores. Here, we construct a mono-disperse mirror-image nanopore, DpPorA and characterise its functional properties. Importantly, we alter the charge pattern and assemble a superior mirror-image pore with enhanced conductance and selectivity under different salt conditions. This pore is used for single-molecule sensing of structurally divergent biomolecules, including peptides, PEGylated polypeptides, full-length alpha-synuclein protein and cyclic sugars. Molecular dynamics simulations confirm these DpPorA are exact mirror-images of LpPorA, further revealing their structurally stable conformation. Fluorescence imaging of giant vesicles reconstituted with mirror-image peptides reveals the formation of large flexible pores facilitating size-dependent molecular transport. To explore biomedical applications, the differential cytotoxic effect of mirror-image peptides and their fluorescently tagged forms on cancer cells demonstrates a significant effect on membrane disruption and cell viability, as opposed to no effect on normal cells. We emphasize that this class of mirror-image pores can advance the development of molecular sensors and therapeutics.

Membrane protein pores are engineered with atomic-level precision for diverse applications in nanotechnology and nanomedicine[1–4]. Among them, natural beta-barrel pores have been extensively studied for single-molecule sensing of nucleic acids, saccharides, polypeptides, proteins, and, more recently, for identification of protein post-translational modifications[5–13]. However, the complexity of engineering and purifying natural pore-forming proteins has spurred interest in synthetic alternatives, such as DNA-based pores[1,13–16]. Despite remarkable progress in assembling sophisticated DNA structures, these nanopores face limitations like electrical leakage, which restrict their utility in nanopore technology[17,18]. De novo-designed beta-barrel pores of smaller conductance and limited molecular selectivity have been constructed[19,20]. Though less explored, transmembrane alpha-helical pores offer significant potential in nanobiotechnology and synthetic biology[1,21–23]. These synthetic pores of low conductance can be created using computational de novo design methods or by mimicking natural alpha-helical assemblies to achieve advanced structural designs and specific functional roles[21–29]. They can be tailored to meet specific requirements, such as size, shape, and charge, for single-molecule sensing due to their structural and functional versatility[23,26,27]. Recently, alpha-helical peptides are gaining importance in targeted therapeutics for cancer therapy and drug delivery[30,31].

Previously, we assembled a functional alpha-helical peptide pore, pPorA, based on the natural membrane porin PorACj[28,32]. Interestingly, the introduction of cysteine in the peptide sequence resulted in the self-assembled preoligomers that formed large stable pores[28]. Further,

[1]Membrane Biology Laboratory, Transdisciplinary Research Program, Rajiv Gandhi Centre for Biotechnology, Thiruvananthapuram, India. [2]Manipal Academy of Higher Education, Manipal, Karnataka, India. [3]School of Science, Constructor University, Bremen, Germany. [4]Microbial Processes and Technology Division, CSIR - National Institute for Interdisciplinary Science and Technology (NIIST), Thiruvananthapuram, India. [5]Academy of Scientific and Innovative Research (AcSIR), Ghaziabad, India. [6]Centre for Human Genetics, Bengaluru, Karnataka, India. [7]These authors contributed equally: Neilah Firzan CA, Kalyanashis Jana, Sreelakshmi Radhakrishnan, Rifat Aara. ✉e-mail: mahendran@rgcb.res.in

we incorporated unnatural D-amino acids into the peptides and demonstrated the formation of stable, functional pores of distinct conductance states, but there was no evidence confirming their mirror-image structure[27]. In this study, we fabricated a stereo-inversed version of pPorA using D-amino acids that form functional mirror-image pores in the lipid membranes. We conducted single-channel electrical recordings to study their functional characteristics and used this pore for single-molecule sensing of various biomacromolecules, including intrinsically disordered amyloid proteins. Molecular dynamics simulations revealed mirror-image pore conformation and molecular transport pathways. Further, we explored the pore-forming properties of these mirror-image peptides and assessed the transport of molecules with varying molecular weights, establishing their functionality in GUVs. Finally, we tested these charge-selective mirror-image pores for their anticancer properties, demonstrating their potential for therapeutic applications.

## Results

### Assembly of stable and functional DpPorA peptide pore

DpPorA peptides containing 40 D-amino acids were chemically synthesized by solid-phase synthesis and purified using HPLC (Fig. 1a and Supplementary Fig. 1). Circular Dichroism spectroscopy of the

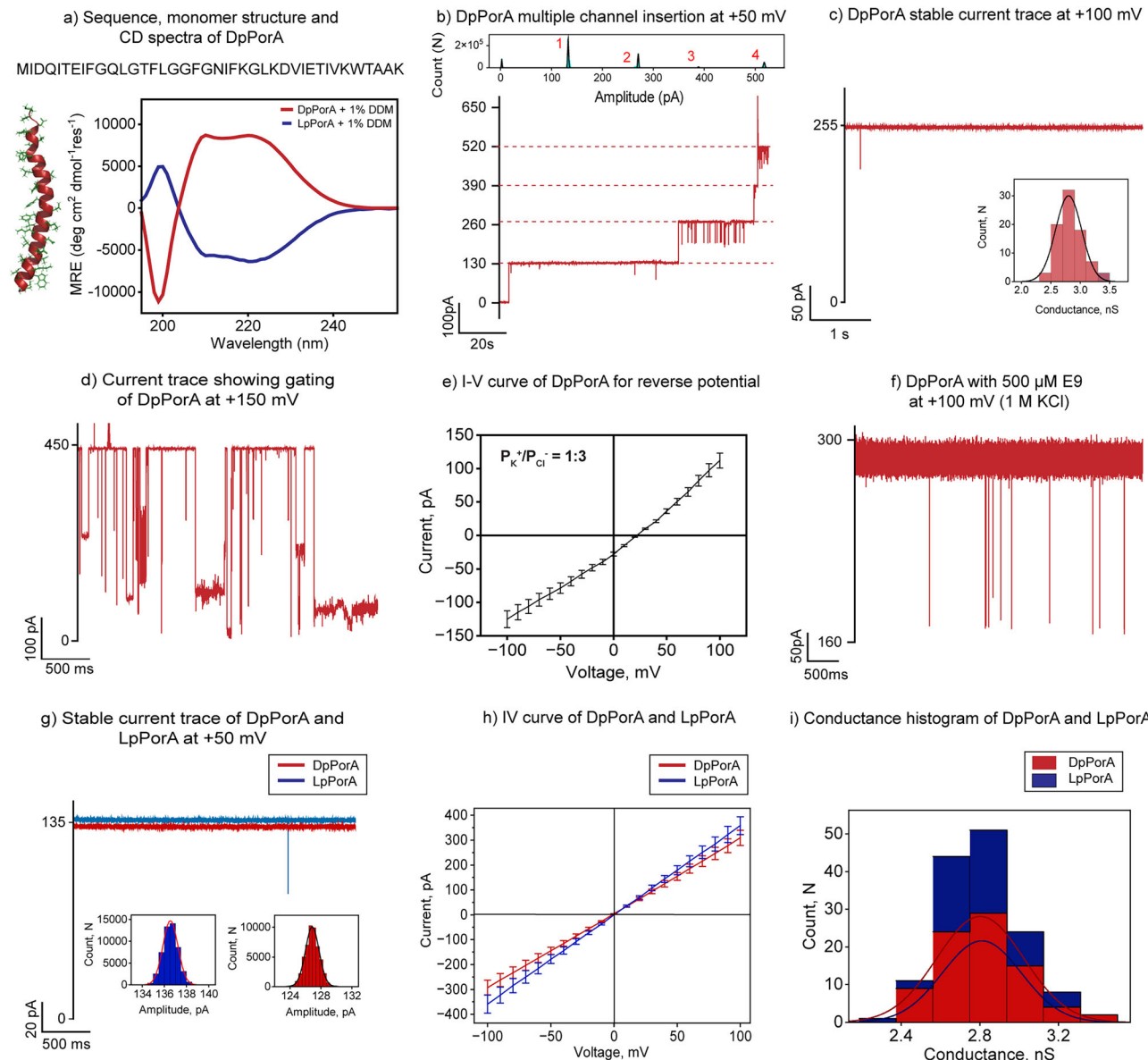

**Fig. 1 | Single-channel characterization and functional stability of DpPorA.**
**a** DpPorA peptide sequence, monomer structure, and CD spectra of DpPorA (red) and LpPorA (blue) in 10 mM phosphate buffer with 1% DDM. **b** Electrical recording of multiple channel insertion at +50 mV with corresponding all-point current amplitude histogram. **c** Stable open conductance trace at +100 mV with unitary conductance histogram of n = 83 insertion events as inset. **d** Electrical recording showing characteristic gating of DpPorA at +150 mV in 1 M KCl buffer. **e** The reverse potential was obtained from the I–V curve of a single DpPorA pore in an asymmetric buffer (0.15 M KCl at cis and 1 M KCl at trans) for charge selectivity. Error bars represent 10% standard error mean between 4 independent experiments.

**f** Interaction of 500 μM E9 with DpPorA on cis side addition at +100 mV in 1 M KCl buffer. **g** Overlapping stable current trace of DpPorA and LpPorA at +50 mV in 1 M KCl buffer. **h** Overlapping I–V curve of DpPorA and LpPorA showing stable current from −100 to +100 mV in 1 M KCl. Error bars represent 10% standard error mean between 4 independent experiments. **i** Stacked mean unitary conductance histogram of DpPorA (red) and LpPorA (blue), obtained by fitting the distribution to Gaussian (number of insertion events, n = 83 and n = 58 for DpPorA and LpPorA, respectively). The current signals were digitally filtered at 2 kHz. The current signals (**f**) were filtered at 10 kHz and sampled at 50 kHz. Electrolyte: 1 M KCl, 10 mM HEPES, pH 7.4.

peptides in 1% n-dodecyl β-D-maltoside (DDM) revealed an alpha-helical conformation with opposite ellipticity to that of L-counterpart pPorA peptides (Fig. 1a and Supplementary Fig. 1). The pore-forming properties of the DpPorA peptides were studied using single-channel electrical recordings. The DpPorA peptides rapidly inserted into 1,2-diphytanoyl-sn-glycero-3-phosphocholine (DPhPC) planar lipid bilayers and formed stable large conductance pores at different voltages (Fig. 1b, c and Supplementary Fig. 2). The statistical analysis of 83 single-channel insertions revealed that these peptides formed uniform pores with a mean unitary conductance (G) of $2.80 \pm 0.2$ nS in 1 M KCl at +50 mV (Fig. 1b, c). The pore remained in the stable open conductance state at the voltages ranging from ±25 mV to ±100 mV and showed gating at voltages above ±150 mV (Fig. 1c, d and Supplementary Fig. 2). Notably, 90% of the pores showed slightly higher current at negative than positive voltages, indicating specific orientation in the membrane (Supplementary Fig. 2). The selectivity measurements suggest that the DpPorA pores preferentially uptake anions with a permeability ratio $P_K^+/P_{Cl}^-$ of 1:3 (Fig. 1e and SI text).

Next, we examined the interaction of the negatively charged polypeptide nonaglutamic acid (E9) with the pores at different analyte concentrations and voltages (Supplementary Fig. 3). The addition of E9 peptides (100 µM and 500 µM) to the cis and trans side of the pore did not produce any significant ion current blockages at voltages ranging from ±25 mV to ±100 mV (Fig. 1f and Supplementary Fig. 3). This data indicates the minimal binding of the peptide analytes with the pore surface most likely due to faster peptide translocation (Fig. 1f and Supplementary Fig. 3). Further, we compared the single-channel electrical properties of LpPorA and DpPorA, revealing that both pores exhibit similar single-channel electrical properties (Fig. 1g–i). More specifically, the L- and D-pores exhibited identical unitary conductance, suggesting that these pores are exact mirror-images of each other (Fig. 1g–i). As per the design, DpPorA peptides remained resistant to proteinase K action and showed a well-defined monomeric band in SDS PAGE, and these peptides readily inserted into DPhPC bilayers, forming stable pores of (G) $2.8 \pm 0.2$ nS in 1 M KCl (n = 25) (Supplementary Fig. 3).

## Electrical and functional properties of DpPorA DE pores

Next, we rationally designed and synthesized a specific peptide, DpPorA DE, in which the negatively charged glutamic acid and aspartic acid at the 28th and 31st positions were replaced with neutral alanine residues to tune the charge-selective molecular transport across the pore (Fig. 2a). Circular Dichroism spectra confirmed the alpha-helical conformation of the DpPorA DE peptide, showing opposite ellipticity compared to that of the LpPorA DE peptide (Fig. 2a and Supplementary Fig. 1).

The multiple and single-channel insertion studies (n = 61) revealed that the DpPorA DE formed uniform pores of a mean unitary conductance of $3.80 \pm 0.2$ nS in 1 M KCl and exhibited gating at higher voltages above ±100 mV (Fig. 2b, c and Supplementary Fig. 4). The selectivity measurements indicated a permeability ratio $P_K^+/P_{Cl}^-$ of 1:20 confirming dominant anion-selective features of the pore (Fig. 2d). Notably, both DpPorA DE and LpPorA DE exhibited identical single-channel conductance, indicating that these pores are mirror-image structures of each other (Fig. 2e, f). Remarkably, these protease-stable DpPorA DE pores exhibit large conductance and high selectivity for anions compared to DpPorA (Fig. 2c, d and Supplementary Fig. 4). Accordingly, we examined the interaction of various negatively charged analytes with the pore surface. We performed all analyte blockage experiments at voltages less than ±50 mV, where no gating was observed and the pore remained in a fully open state (Supplementary Figs. 5 and 6). Adding 100 µM E9 to the cis side of the pore resulted in ion current blockages at positive voltages, whereas no blockages were observed at the negative voltages, indicating electrostatic binding of the E9 peptides with the pore lumen (Fig. 2g and Supplementary Fig. 5). The mean dwell time of blockage ($\tau_{off}$) was

calculated to be $0.2 \pm 0.02$ ms at +20 mV, which decreased with an increase in the voltage corresponding to an increase in the dissociation rate ($k_{off}$), establishing the successful peptide translocation (Fig. 2g and Supplementary Fig. 5). Interestingly, at +40 mV, E9-induced ion current blockage events were short less frequent with $\tau_{off}$ of $0.1 \pm 0.1$ ms that are not fully resolved demonstrating the rapid peptide translocation events (Fig. 2g). Comparing the interaction with the wild-type DpPorA, removing negatively charged residues affects the overall charge distribution on the DpPorA DE pore, enhancing the binding affinity of E9 peptides to the pore surface facilitating the peptide binding and transport (Fig. 1f, g). Interestingly, the addition of 100 µM E9 to the trans side produced more dominant ion current blockages compared to the cis side addition indicating an asymmetry in the peptide binding (Fig. 2g and Supplementary Fig. 5). The substitution of negatively charged residues towards the C-terminal (trans side) has created a positive surface charge, attracting the negatively charged E9 to bind more effectively to the trans side, further justifying the pore design (Fig. 2g and Supplementary Fig. 5).

Next, we explored the ability of the DpPorA DE to sense larger complex analytes, such as PEGylated peptides, which mimic the characteristics of intrinsically disordered proteins[4]. To achieve this, we conjugated E9 peptides with water-soluble polyethylene glycol (PEGs) polymers of molecular weight 200 Da and characterized the binding events (Fig. 2h). Adding 500 µM of PEG 200-E9 peptides to the cis side of the pore produced well-defined ion current blockages at different positive voltages confirming electrostatic peptide binding (Fig. 2h and Supplementary Fig. 5). The number of blockage events increased with an increase in the applied voltage indicating the need for a high voltage force to pull the analytes through DpPorA. These electrophoretic pulling of PEG 200-E9 peptides into the pore resulted in strong binding with the $\tau_{off}$ calculated to be $0.35 \pm 0.03$ ms at +40 mV (Fig. 2h and Supplementary Fig. 5). Similar binding patterns were observed for the trans-side addition of the PEGylated E9 peptides (Supplementary Fig. 5). Our data show that the PEGylated E9 peptides bind to the pore with high affinity ($K_D = 0.5 \times 10^{-3}$ M) at +40 mV compared to E9 peptides ($K_D = 3 \times 10^{-3}$ M) due to its large unstructured conformation. Notably, adding cationic PEGylated nonaarginine peptides did not produce ion current blockages, confirming the charge-selective translocation of anionic molecules (Supplementary Fig. 5).

Further, we explored the interaction of a natural intrinsically disordered protein, alpha-synuclein (αS), with mirror-image nanopores. Notably, alpha-synuclein (αS) aggregation results in Parkinson's disease, and these αS aggregates can adopt various conformations, making their detection difficult[33]. Adding 25 nM αS to the cis side of the pore produced well-defined ion current blockages at positive voltages, suggesting electrophoretic pulling and electrostatic binding (Fig. 2i and Supplementary Fig. 5). In line with this, the blockage events increased with the voltage, and the pore exhibited three different blocking states, each corresponding to different protein conformations due to structural destabilization (Fig. 2i and Supplementary Fig. 5). Our experimental data highlight the charge-selective binding of negatively charged αS with the anion-selective mirror-image pores. These results suggest the potential of large flexible mirror-image nanopores over solid-state nanopores, enabling label-free single-molecule detection with enhanced sensitivity[33]. Next, we performed competitive binding experiments of E9 peptides, PEGylated E9 peptides, and alpha-synuclein with a DpPorA DE pore. Notably, E9 peptides induced short ion current blockages of ~100 µs, while PEGylated E9 peptides produced events of ~300 µs. Subsequent addition of alpha-synuclein produced numerous, well-defined, 3-step distinct ion current blockages (Fig. 2j). These data show that each analyte produced mutually exclusive current blockages, confirming selective competitive binding with the pore, demonstrating the structural compatibility and functional advantage of these pores for advanced nanopore sensing. Finally, we examined the interaction of bulky cyclic sugars that are structurally distinct from the tested peptide

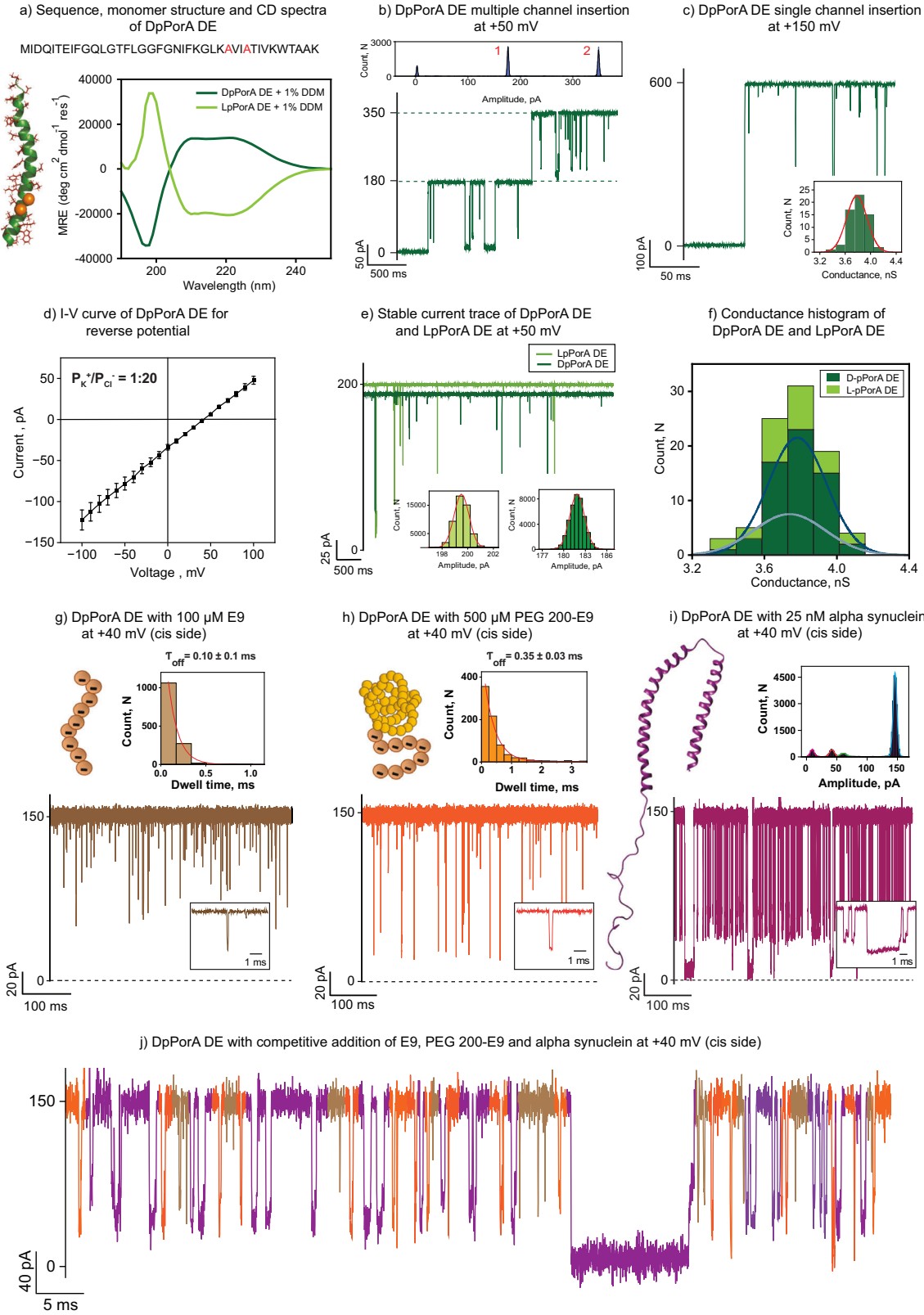

a) Sequence, monomer structure and CD spectra of DpPorA DE

MIDQITEIFGQLGTFLGGFGNIFKGLKAVIATIVKWTAAK

b) DpPorA DE multiple channel insertion at +50 mV

c) DpPorA DE single channel insertion at +150 mV

d) I-V curve of DpPorA DE for reverse potential

$P_K^+/P_{Cl}^- = 1:20$

e) Stable current trace of DpPorA DE and LpPorA DE at +50 mV

f) Conductance histogram of DpPorA DE and LpPorA DE

g) DpPorA DE with 100 µM E9 at +40 mV (cis side)

$\tau_{off} = 0.10 \pm 0.1$ ms

h) DpPorA DE with 500 µM PEG 200-E9 at +40 mV (cis side)

$\tau_{off} = 0.35 \pm 0.03$ ms

i) DpPorA DE with 25 nM alpha synuclein at +40 mV (cis side)

j) DpPorA DE with competitive addition of E9, PEG 200-E9 and alpha synuclein at +40 mV (cis side)

analytes with the mirror-image pores. Adding 100 µM anionic six-fold symmetric alpha cyclodextrin ($s_6\alpha$CD) and eight-fold symmetric gamma cyclodextrin ($s_8\gamma$CD) resulted in well-defined ion current blockages at positive voltages, indicating rapid translocation of CDs in line with the electrophoretic pulling (Supplementary Fig. 5). The ability of the mirror-image pores to detect various analytes of distinct conformation including biologically relevant proteins establish their functional versatility, thus making it an efficient nanopore for single-molecule sensing. We also studied the single-channel properties of the pore in the presence of low salt buffer (0.15 M KCl) and divalent ions (1 M MgCl$_2$) (Supplementary Fig. 7). The stable pores were formed under different salt conditions and DDM detergent concentrations, establishing the structural integrity of the pore for single-molecule sensing and cell-based assays.

**Fig. 2 | Single-channel characterization and functional stability of DpPorA DE.**
**a** DpPorA DE peptide sequence, monomer structure, and CD spectra of DpPorA DE (dark green) and LpPorA DE (light green) in 10 mM phosphate buffer with 1% DDM. **b** Electrical recording of multiple channel insertion at +50 mV with corresponding all-point current amplitude histogram. **c** Single-channel insertion at +150 mV with unitary conductance histogram of n = 61 insertion events, as inset. **d** The reverse potential was obtained from the I–V curve of a single DpPorA DE pore in an asymmetric buffer (0.15 M KCl at cis and 1 M KCl at trans) for charge selectivity. Error bars represent 10% standard error mean between 4 independent experiments. **e** Overlapping stable current trace of DpPorA DE and LpPorA DE at +50 mV. **f** Stacked mean unitary conductance histogram of DpPorA DE (dark green) and LpPorA DE (light green), obtained by fitting the distribution to Gaussian (number of

insertion events, n = 61 and n = 26 for DpPorA DE and LpPorA DE, respectively). **g** Interaction of 100 μM E9 with DpPorA DE on the cis side at +40 mV. **h** Interaction of 500 μM PEG 200-E9 with DpPorA DE on the cis side at +40 mV. Insets show corresponding dissociation dwell time ($\tau_{off}$) histogram fitted with a mono-exponential probability function and recording at an expanded time scale. **i** Interaction of 25 nM alpha-synuclein (αS) with DpPorA DE on the cis side at +40 mV. Insets show corresponding current amplitude histogram and recording at an expanded time scale. **j** Competitive interaction of 100 μM E9 (brown), 500 μM PEG 200-E9 (orange), and 25 nM alpha-synuclein (αS) (magenta) with DpPorA DE (cis side) at +40 mV. The current signals (**b**, **c**, **e**) were digitally filtered at 2 kHz. The current signals (**g–j**) were filtered at 10 kHz and sampled at 50 kHz. Electrolyte: 1 M KCl, 10 mM HEPES, pH 7.4.

## Molecular dynamics simulations unveiling pore characteristics

The model pores LpPorA, DpPorA, LpPorA DE, and DpPorA DE were constructed using the CCBuilder web server as starting points for the molecular dynamics (MD) simulations (Fig. 3a, b)[34]. The possibility of the formation of octameric and hexameric pores by the given sequences was explored, and the hexameric pores were unstable in applied electric field simulations. Accordingly, all simulations described below were performed using stable octameric pores.

The Ramachandran plots of the different pores indicated that the D-pores are mirror-images of the L-pores (Supplementary Fig. 8 and SI text). The Root Mean Square Deviations (RMSDs) analysis shows notable fluctuations in the RMSD values during the initial 100 ns, followed by a flattening of the curves (Supplementary Figs. 9 and 10). The resulting structures from the 500 ns-long unbiased equilibration runs were used as starting structures in applied electric field MD simulations. The designed pores were stable during these applied field simulations, as can be seen by their RMSD values calculated with respect to the final structure from the respective unbiased simulation (Supplementary Figs. 9 and 10). A transmembrane potential was applied in the simulations using the constant electric field approach to investigate the potassium and chloride ion permeability and ionic conductance of the pores. At an applied voltage of +0.5 V, the simulations resulted in similar conductance values of ~3.5 nS for LpPorA and DpPorA, confirming that these pores are mirror-images of each other as the permeant ion is not chiral (Supplementary Table 1). Also, the ion conductance at −0.5 V showed increased fluctuations of the flexible C-terminal amino acids, resulting in slightly higher conductance (Supplementary Table 1). Furthermore, the ratio of the obtained Cl⁻ and K⁺ currents, i.e., I⁻/I⁺, indicates an anion selectivity of the pores. The uneven distribution of charged amino acid residues with a prevalence of positively charged residues on the interior surface at the C-terminal side resulted in the anion selectivity (Supplementary Fig. 11). The analysis of the average radius profile using 100 channel structures from the 500 ns unbiased simulation performed using the HOLE program unveiled that both the L- and D-pores exhibit a minimum radius of approximately 6.0 Å (Fig. 3c). Next, we calculated the ion permeability and ionic conductance of the mutant LpPorA DE and DpPorA DE pores, which showed similar conductance values (Supplementary Table 1). Interestingly, these pores showed higher ion conductance and an increased anion-cation permeability ratio compared to wild-type LpPorA and DpPorA pores, which agreed with experimental results (Supplementary Table 1). We attribute this to dominant positive charges at the inner surface of the DpPorA DE pores, resulting in the asymmetric charge distribution that contributes to the anion selectivity and higher conductance (Fig. 3d, e, Supplementary Figs. 12 and 13). The average radius profiles, calculated using the HOLE program, reveal a minimum radius of approximately 5.0 Å for both the L and D mutant pores (Fig. 3f). For a rough estimate of the conductance, we assumed a cylindrical double-cone structure of the pore. Using rough values of 30 Å for the wider diameter, 10 Å for the diameter in the constriction zone, a pore length of 80 Å, and a specific conductivity for the electrolyte KCl of 1.1 nS/Å, one reaches a

conductivity of approximately 3.25 nS[27,35,36]. Additionally, the MD-derived pore radius profiles show that the position of the channel constriction (the "gate") shifts along the Z-axis in opposite directions for the wild and the mutant pores (Supplementary Fig. 14). This shift arises due to asymmetries in the molecular architecture of the pores—specifically, the removal of acidic amino acid residues in the mutant pores DpPorA DE and LpPorA DE shift the gate in opposite directions with respect to that of the wild-type DpPorA and LpPorA. In the wild-type DpPorA and LpPorA pores, steric or electrostatic constraints lead to a central narrowing toward negative Z, while the mutant DpPorA DE and LpPorA DE pores show a shift of the gate toward positive Z values (Supplementary Fig. 14). Despite the smaller radius, these mutant DpPorA DE pores show higher conductance due to the different charges and a larger charge separation on the inner surface (Fig. 3g–i and Supplementary Fig. 15). Accordingly, we analyzed the electrostatic potential maps to elucidate the charge distribution across the inner and outer surfaces of the DpPorA and DpPorA DE pores. Our data show that the DpPorA DE pores exhibit a more positive electrostatic potential surface along the pore lumen compared to DpPorA due to the lysine residues (Fig. 3g–i and Supplementary Fig. 15). The positional asymmetry influenced molecular transport by altering the local potential energy landscape and the hydration environment, resulting in the slightly different conductance values of these pores.

Furthermore, unbiased MD simulations revealed the molecular binding of E9 peptides with DpPorA and DpPorA DE, where a stabilization was observed through hydrogen bonding and electrostatic interactions with the positively charged lysine residues in the pore (Fig. 4a and Supplementary Fig. 16).

To examine the translocation dynamics of E9 peptides through both the DpPorA and DpPorA DE pores, steered MD simulations (SMD) were carried out to estimate the forces necessary to facilitate the transport (Fig. 4a and Supplementary Fig. 16). The average maximum force needed to pull E9 through DpPorA was approximately 210 pN and around 200 pN through DpPorA DE, i.e., showing no statistically significant differences in these simulations (Fig. 4 and Supplementary Fig. 16). Notably, SMD simulations were performed by maintaining a small constant pulling velocity of 0.1 Å/ns in attempt to keep the artificial distortions small. As seen in Fig. 4, the maximum force was necessary in the constriction region of the pore. The structural analysis showed that E9 maintained a flexible, unstructured conformation with only one end involved in electrostatic interactions with lysine residues, promoting the charge-selective peptide translocation through the DpPorA DE pore (Fig. 4a). Although the pulling speed might lead to forces that are too high in absolute numbers, the force profile as a function of the channel axis nevertheless contains important information on the translocation process. Moreover, many experimental studies using atomic force microscopy (AFM) have reported that unfolding individual protein domains, such as titin immunoglobulin domains, typically requires forces in the range of 150–300 pN, depending on the pulling rate and protein structure. Thus, the ~200 pN pulling forces observed in our MD simulations fall within the biologically realistic regime for mechanical manipulation of protein or

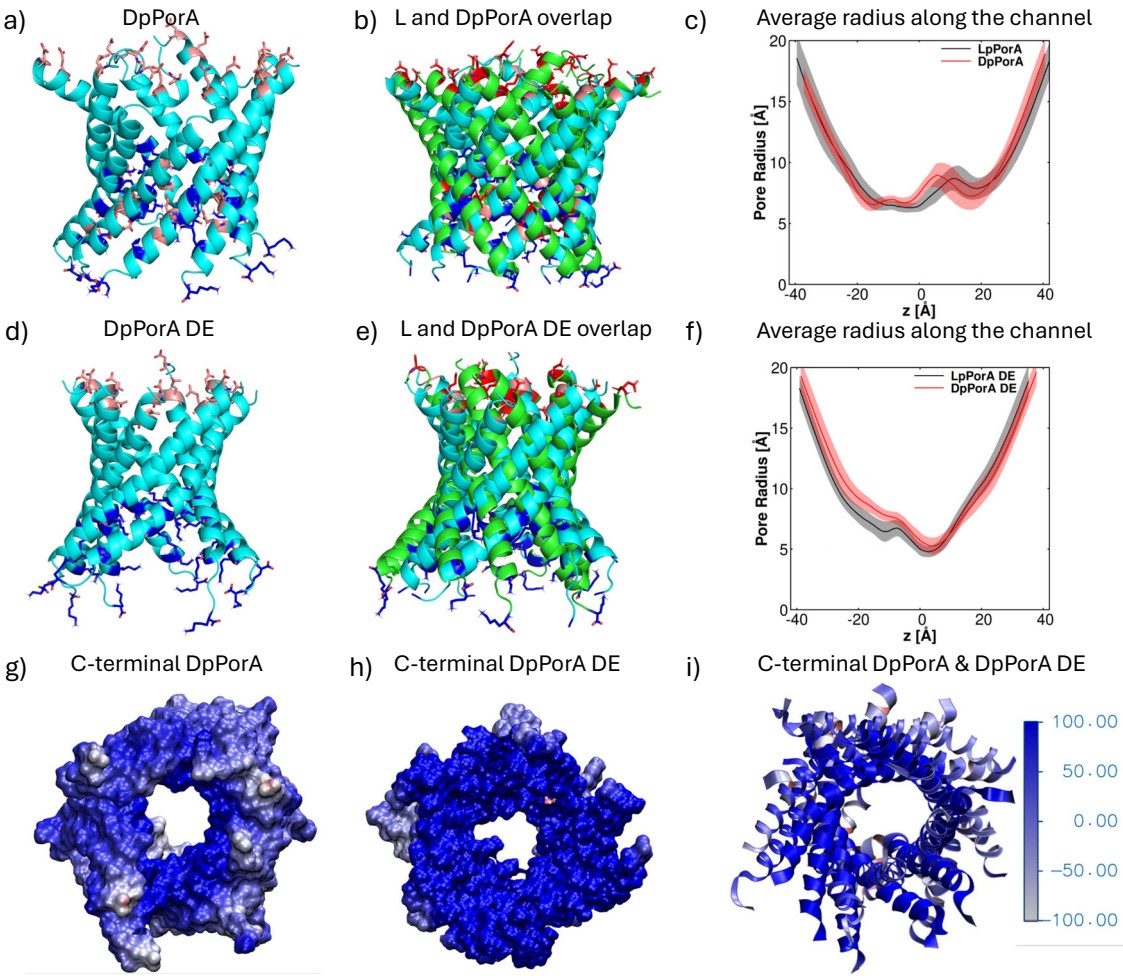

**Fig. 3 | Structures and radius profiles along the channel axis of the designed pores. a** Stable DpPorA (cyan) structures are shown in cartoon representation. Basic amino acid residues are highlighted in blue, while acidic residues are depicted in red. **b** Stable LpPorA (green) and DpPorA (cyan) structures are overlapped in cartoon representation. Basic amino acid residues are highlighted in blue, while acidic residues are depicted in red. **c** Average radius profiles along the channel axis are based on unbiased trajectories, with the transparent shades indicating standard deviations in the average radius calculations. **d** Cartoon representations of stable DpPorA DE (cyan) structures of the mutated pPorA. **e** Overlapped cartoon representations of stable LpPorA DE (green) and DpPorA DE (cyan) structures of the mutated pPorA. **f** The average radius profiles of the mutated pores along the channel axis from a trajectory, while the transparent shades indicating standard deviations in the average radius calculations. **g** C-terminal views of the electrostatic potential maps highlight the charged amino acid residues of DpPorA lining the interior channel wall. The computed electrostatic potential ranges from −188 to +309 $k_BT/e$ for DpPorA, where 1 $k_BT/e$ equals 26 mV at 300 K. To enhance clarity, the color range for the displayed electrostatic potential maps has been restricted to a range from −100 to +100 $k_BT/e$. **h** C-terminal electrostatic potential map of DpPorA DE, where the computed electrostatic potential ranges from −299 to +775 $k_BT/e$ for DpPorA DE. **i** The overlapped representation of the C-terminal electrostatic potential map of DpPorA (displayed with a narrower ribbon) and DpPorA DE (displayed with a broader ribbon).

peptide backbones[37–39]. Furthermore, the translocation pathway of PEG 200-E9 through the DpPorA and DpPorA DE pores was explored (Fig. 4b and Supplementary Fig. 16). The force required to pull the PEG 200-E9 through DpPorA and DpPorA DE was approximately 460 pN. The translocation mechanism of E9 and PEG 200-E9 was similar, but the latter required higher force due to its larger size and additional interactions governed by the N-terminal PEGylation of E9. The peak of the force profile for E9 translocation was observed in the middle of the translocation pathway, whereas for the PEG 200-E9 translocation, two humps were observed in the force profile due to the interaction of the PEG moiety and the E9 moiety (Fig. 4b and Supplementary Fig. 16).

### Assembly and transport across mirror-image pores in giant unilamellar vesicle system

Single-channel and MD simulations data confirmed the formation of structurally stable mirror-image DpPorA pores. Next, we examined specifically the pore-forming activity of DpPorA and DpPorA DE in alternative membrane models such as giant unilamellar vesicles (GUVs) (Fig. 5). This will allow us to understand the assembly mechanism of the pore formation, which will subsequently help design sophisticated pores for synthetic nanobiotechnology applications. The size-dependent permeabilization of different hydrophilic fluorophores was determined (Alexa Fluor 350 Hydrazide: M.W-349 Da, Alexa Fluor 555 Hydrazide: M.W-1150 Da, ATTO 488 Dextran: M.W-3000 Da) (Fig. 5). The hydrophilic molecules did not diffuse across the control GUVs (with 0.1% DDM alone) (Fig. 5a). Moreover, the diffusion of the fluorophores in GUVs reconstituted with DpPorA and DpPorA DE peptide was quantified (Fig. 5b, c). Fluorophore uptake was examined as time dependence to quantify the rate at which the influx of Alexa 350 hydrazide occurs. In peptide-reconstituted GUVs, the $I_{in}/I_{out}$ value increased with time (in minutes) (Fig. 5d, e). In contrast, the control vesicles incubated with 0.1% DDM remained impermeable to the fluorophores even after 45 min (Fig. 5). Statistical analysis was performed after the time-dependent study to quantify the percentage of

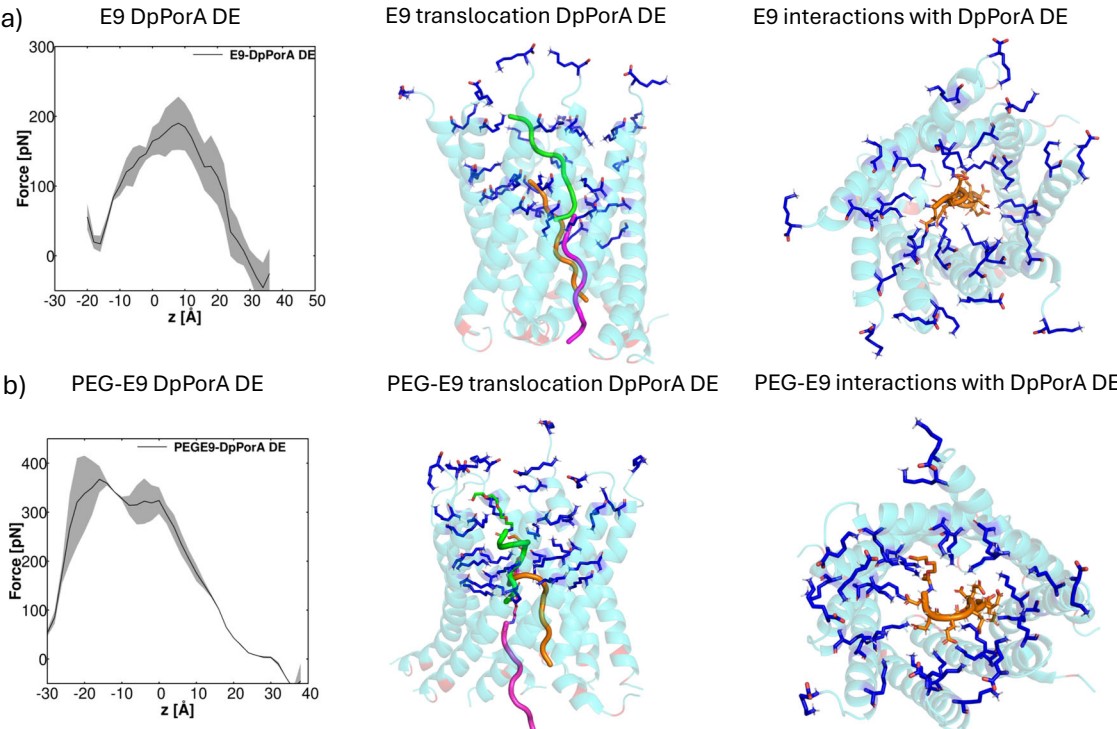

**Fig. 4 | Translocation of E9 and PEG 200-E9 through DpPorA DE. a** Average force profile from steered MD simulations depicting the permeation of anionic nonaglutamic acid as a function of reaction coordinate $z$. The average force profile is depicted as a bold line, and the standard deviation as a transparent shade. This coordinate corresponds to the center of the mass distance between nonaglutamic acid and the $C_\alpha$ atoms of DpPorA DE. The shaded error bars represent the standard deviations from the three simulations. In the middle panel, three different positions of E9 in the DpPorA DE are shown from the translocation pathway: the initial

position (green), one in the middle of the pore corresponding to the peak position of the force profile (brown), and the final conformation (magenta) showing the E9 peptide leaving the interaction zone of DpPorA DE. The interactions between the negatively charged acidic groups of the E9 peptide and the NH3+ groups of the lysine residues along the translocation pathway through DpPorA DE are shown in the right part. **b** Same as (**a**), but for PEG 200-nonaglutamic acid as substrate. The PEG 200 moieties are shown as stick structures attached to the nonaglutamic acid peptide.

vesicles permeable to the fluorophores (details in methods). When a peptide concentration ranging from 10 to 15 μM was used in DOPC vesicles, the percentage of permeabilization of Alexa Fluor 350, Alexa Fluor 555, and ATTO 488 dextran was observed to be 88.9 ± 7.8, 67.9 ± 10.4, and 73.0 ± 14.1, respectively, for DpPorA (Fig. 5f).

Vesicles, when reconstituted DpPorA DE showed a similar % of permeabilization with 89.2 ± 3.7 (Alexa Fluor 350), 71.3 ± 3.5 (Alexa Fluor 555), and 53.8 ± 8 (ATTO 488 Dextran) (Fig. 5g). Control vesicle permeabilization was below 15% for all molecules. Furthermore, mirror-images of lipid DOPC (Cis and Enantio DOPC) were used to make vesicles to study the functional activity of mirror-image DpPorA pores[40,41]. Both Cis-DOPC and Ent-DOPC lipid configurations supported effective peptide pore assembly, and comparable membrane permeabilization was observed for DpPorA across all tested fluorophores (Supplementary Fig. 17). This demonstrates stable pore formation in enantiomeric lipids in giant unilamellar vesicle systems, which is challenging for other membrane platforms[41]. Our data confirms that the DpPorA and DpPorA DE form large pores, facilitating the uptake of molecules up to 3 kDa into the vesicles, which agrees with single-channel electrical recordings and MD simulations data.

### Effect of DpPorA and DpPorA DE peptides on cancer cells

The insights gained into the molecular basis of pore formation by mirror-image peptides give us a unique opportunity to test their biological activity in complex mammalian cell membrane systems as an anticancer strategy. We have shown that incorporating the unnatural D-amino acid has altered the stereochemistry and overall spatial arrangement, rendering the DpPorA peptides resistant to protease action (Supplementary Figs. 3 and 4). Accordingly, we tested the effect

of the DpPorA and LpPorA peptides on the MDA-MB-231 human cancer cells, which represent the Triple Negative Breast Cancer (TNBC) subtype. No targeted therapy is available for this subtype, leading to a poor prognosis for patients. Therefore, we propose targeting cancer cells with DpPorA and LpPorA peptides via membrane-specific interactions and pore formation.

We examined the solubility and folding of peptides in different DDM concentrations by size exclusion chromatography and evaluated the cytotoxic effects of the DDM on MDA-MB-231 cells to confirm the safe concentration range for subsequent experiments (Supplementary Fig. 18 and Supplementary Table 2). The cells were treated with DDM concentrations ranging from 0.001% to 0.1% for 24 h, following which cell viability was assessed using the MTT assay (Supplementary Fig. 18). DDM concentrations up to 0.007% did not have a significant cytotoxic effect and were comparable to the Control. We determined the effect of the DpPorA and LpPorA peptides on the MDA-MB-231 cancer cells using the MTT assay. There was no significant change in viability when cells were treated with 10 μM and 20 μM DpPorA and LpPorA (Supplementary Fig. 18, Supplementary Table 3). We then analyzed the effect of 10 μM and 20 μM DpPorA DE and LpPorA DE on the cell viability. DpPorA DE peptide had a greater effect on cell viability when compared to LpPorA DE (Supplementary Fig. 19 and Supplementary Table 3). The effect of the DpPorA DE peptide was more pronounced on cancer cell viability when we increased the concentration to 25 μM (Fig. 6a, Supplementary Fig. 19 and Supplementary Table 3). At 25 μM DpPorA DE concentration, a significant reduction of 27.21% in cell viability was observed compared to the Control and DDM-treated groups. The cytotoxic effect is concentration-dependent, as we observed 11%, 19% and 27.21% reduction in cell viability at 10 μM, 20 μM,

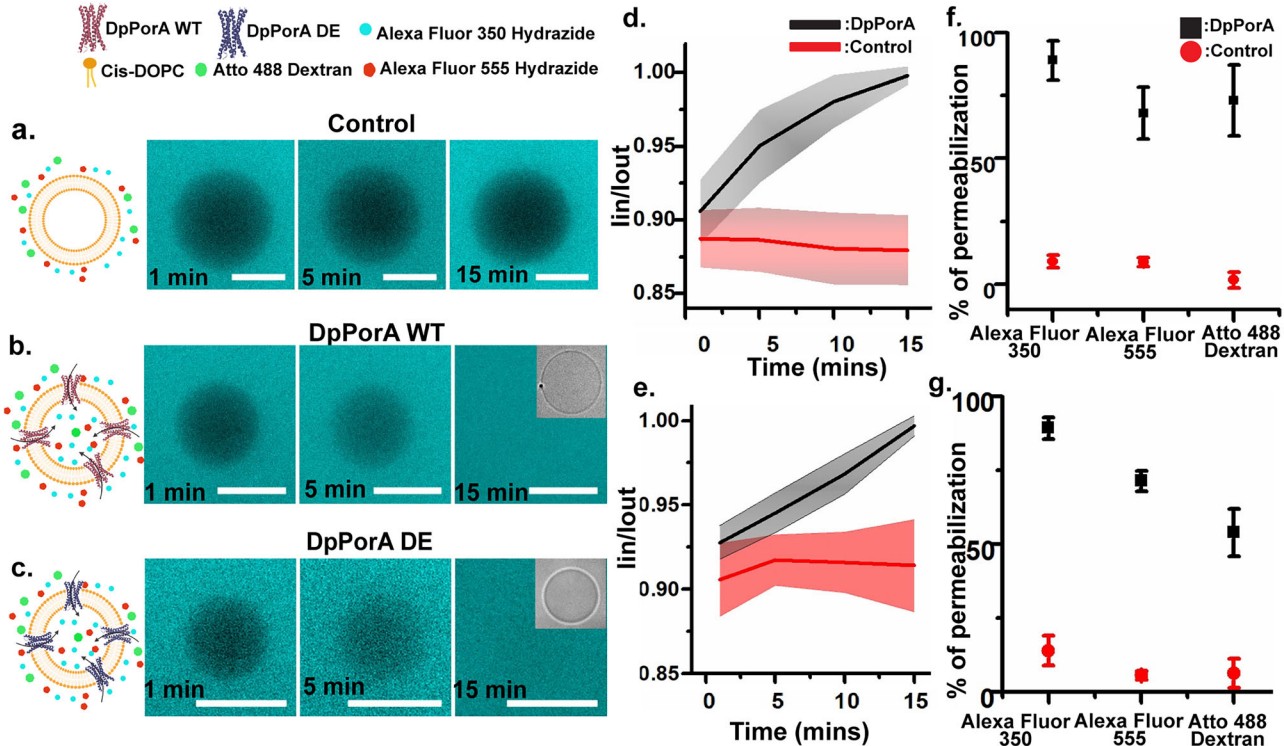

**Fig. 5 | Transport across DpPorA and DpPorA DE pore in giant unilamellar vesicle system.** Schematic and representative images depicting the transport of Alexa Fluor 350 in **a** Control, **b** DpPorA, and **c** DpPorA DE. Graph showing the time-dependent analysis of normalized intensities of individual vesicles in the presence of **d** DpPorA (n = 8 vesicles each) and **e** DpPorA DE (n = 10 vesicles each). Statistical analysis of the percentage of permeabilization of hydrophilic fluorescent dyes in **f** DpPorA incorporated vesicles (n = 136, 64, and 72 individual vesicles for Alexa Fluor 350, Alexa Fluor 555, and ATTO 488 Dextran, respectively from N = 4 batches) and control vesicles (n = 123, 63, and 60 vesicles for Alexa Fluor 350, Alexa Fluor

555, and ATTO 488 Dextran, respectively from N = 4 batches). **g** DpPorA DE incorporated vesicles (n = 125, 62, and 63 vesicles for Alexa Fluor 350, Alexa Fluor 555, and ATTO 488 Dextran, respectively, from N = 4 batches) and control vesicles (n = 143,70 and 73 vesicles for Alexa Fluor 350, Alexa Fluor 555, and ATTO 488 Dextran, respectively, from N = 4 batches). The percentage of permeabilization values was denoted as mean ± S.D. The number of batches and the total number of vesicles are denoted as "N" and "n," respectively. Buffer conditions: 100 mM KCl in 10 mM HEPES (pH: 7.4). Scale bar: 10 μm. Schematic figures of GUVs are created with BioRender.com.

and 25 μM DpPorA DE peptide concentrations, respectively. These results suggest that the protease-stable DpPorA DE peptide, due to its dominant positive charge, interacts with a negatively charged complex cancer cell lipid bilayer, allowing pore formation and leading to cytotoxic effects. Interestingly, 20 μM DpPorA DE had no significant effect on the MCF10A cells, representing the normal mammary cells (Supplementary Fig. 19). Next, we looked at the effect of DpPorA DE peptides on the integrity of the cell membrane using immunofluorescence staining with CellMask Deep Red, which labels the plasma membrane. We observed two different phenotypes based on cell membrane integrity: Intact and Disrupted (Supplementary Fig. 20). Based on the phenotypes observed, we binned cells into these two categories and counted at least 100 individual cells in Control, 0.00125% DDM-treated, and 25 μM DpPorA DE-treated cells. The Control and DDM cells showed a mixture of both phenotypes (Supplementary Table 4). We observed a distribution of 67.38% and 32.62% for Control and 71.3% and 28.7% for DDM, with Intact and Disrupted phenotypes, respectively. Remarkably, we observed a significant increase in the Disrupted phenotype in the DpPorA DE peptide-treated cells compared to the Control (99.37% for DpPorA DE versus 32.62% for Control) (Fig. 6b–d and Supplementary Table 4). This result suggests that the DpPorA DE peptide can disrupt the integrity of the cell membrane of cancer cells, which is reflected in the significant decrease in their viability.

Finally, we examined the pore-forming property of the DpPorA DE peptide tagged with 5-carboxyfluorescein (5-FAM) in planar lipid bilayers and mammalian cell membranes (Fig. 6, Supplementary Figs. 21 and 22 and SI text). This 5-FAM peptide rapidly inserted into the DPhPC bilayers and formed stable, well-defined pores at different

voltages, similar to untagged peptides in single-channel recordings (Supplementary Fig. 21). We next quantified the incorporation of 5-FAM-DpPorA peptides into the living cancer cell membrane (MDA-MB-231 cells). These peptides incorporated into the cell membrane within 4 h, as visualized by the green channel in the cell membrane, with increasing intensity observed by 24 h (Fig. 6e, f and Supplementary Fig. 22). For example, a single-cell fluorescence intensity scatter plot confirms time-dependent peptide incorporation in the cell population.

## Discussion

Engineering channels that conduct ions or molecules selectively could lead to nano biosensors for applications in nanotechnology and medicine, particularly for sensing and sequencing biomacromolecules[1–3,5–13,42]. Here, we designed and assembled a membrane-spanning mirror-image pore composed of stereo-inversed D-amino acids adapted from a natural bacterial channel. We emphasize that these functional mirror-image pores are structurally stable and exhibited unusually high conductance compared to other synthetic pores reported to date, thus advantageous for single-molecule sensing of structurally distinct analytes, including pathogenic alpha-synuclein[4,33]. These flexible pores can be easily incorporated into solid-state devices to develop portable nanopore sensors that will open up new opportunities in medical diagnosis[16,43].

Previously, we have demonstrated the formation of pores composed of D-amino acid peptides, but there has been no data to establish mirror-image pore structures as L- and D-pores showed distinct single-channel electrical and functional properties[27,28,40]. We attribute this to the specific cysteine in the sequence driving the

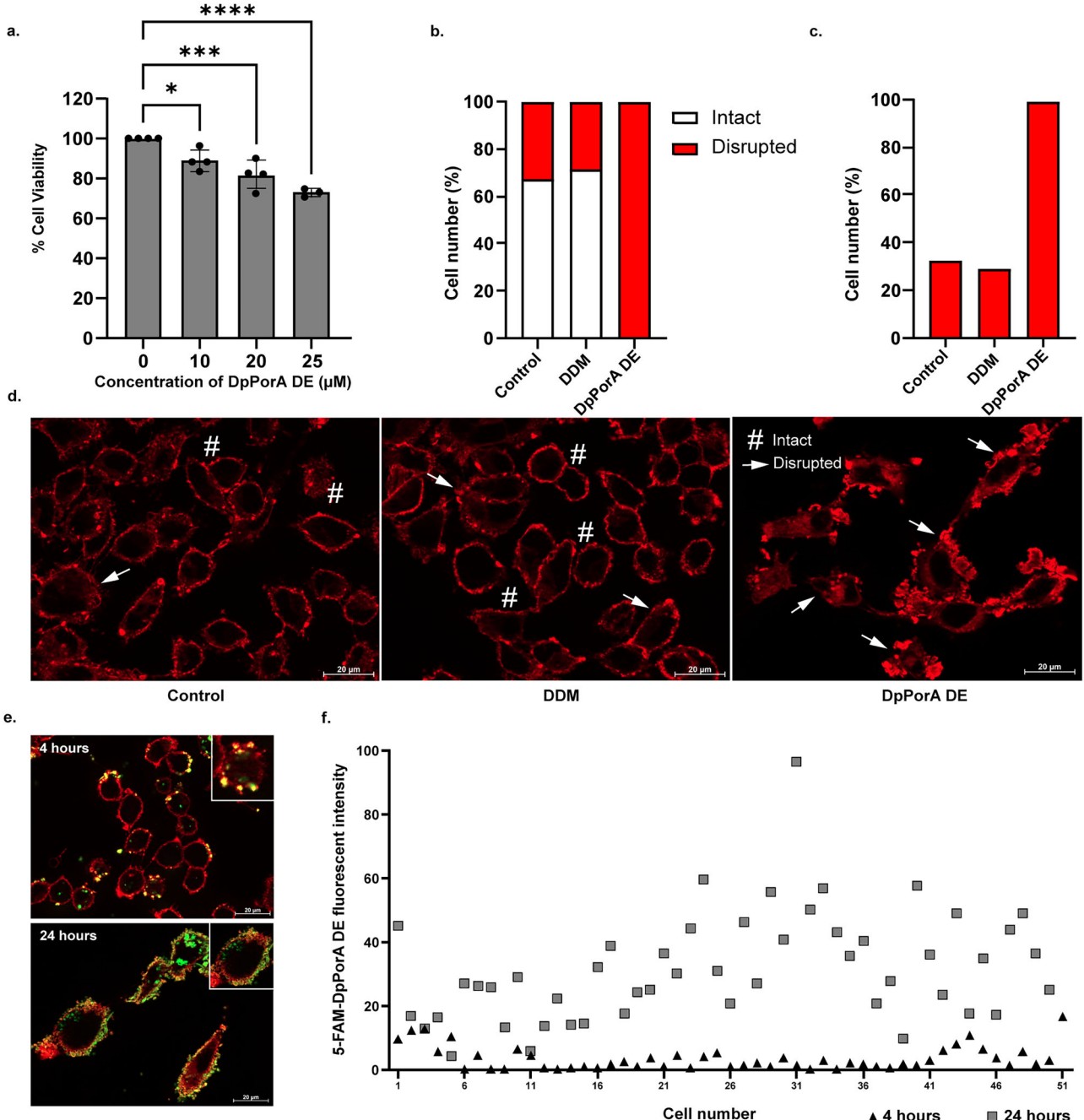

**Fig. 6 | Effect of DpPorA DE peptide on MDA-MB-231 cells. a** Cell viability was assessed by MTT assay 24 h post-treatment with 10 μM, 20 μM, and 25 μM DpPorA DE peptide. Data represents mean ± SEM from n = 4 (0 μM, 10 μM and 20 μM) and n = 3 (25 μM). Each dot represents a biological replicate, where each replicate was an independently seeded and treated culture of MDA-MB-231 cells on different days with different passage numbers. Statistical analysis was performed using one-way ANOVA with Dunnett's multiple comparisons test comparing each treatment to the untreated control (0 μM); Adjusted p values correspond to *p = 0.0206 (10 μM), ***p = 0.0006 (20 μM), ****p < 0.0001 (25 μM) vs. 0 μM. **b** Immunofluorescence staining with CellMask Deep Red was used to evaluate the integrity of the cell membrane in Control, 0.00125% DDM and 25 μM DpPorA DE-treated cells. Cells were categorized into two phenotypes based on membrane integrity: Intact and Disrupted. Control cells displayed a mixture of phenotypes with a distribution of 67.38 % Intact and 32.62 % Disrupted. **c** DpPorA DE-treated cells showed a significant increase in the Disrupted phenotype, with 99.3% of cells being Disrupted, indicating a substantial impact of the DpPorA DE peptide on cell membrane integrity. **d** Fluorescence microscopy images showing different cell membrane integrity phenotypes in Control, 0.00125% DDM, and 25 μM DpPorA DE peptide-treated cells. The two different phenotypes are represented as a hashtag for Intact and the arrow indicates Disrupted cells (100 cells per group, scale bar 20 μm, magnification 63×). **e** Representative fluorescence images showing the distribution of 5-FAM-DpPorA DE (green) in MDA-MB-231 cells at 4 h and 24 h post-addition of peptide. Cell membranes were stained with CellMask (red) (50 cells per group, scale bar 20 μm, magnification 63×). **f** Single-cell fluorescence intensity scatter plot showing increased incorporation of 5-FAM-DpPorA DE at 24 h compared to 4 h, indicating time-dependent peptide incorporation in the cell population.

self-assembly of peptide monomers into octameric preoligomers before membrane insertion. Interestingly, the DpPorA and DpPorA DE demonstrated in this study were devoid of cysteine residues that formed mirror-image pores only in the membranes and exhibited similar pore conductance to their L counterparts, as permeant ions are not chiral (Figs. 1i and 2f)[40]. Importantly, simple site-specific charge modification of these peptides resulted in a significant increase in ion conductance and selectivity of functional mirror-image pores that is

challenging to obtain by redesigning the open beta-barrel structures[1,2,11]. These mirror-image pores also showed high stability to protease and formed functional pores in the lipid bilayers (Supplementary Figs. 3 and 4). MD simulation data established a similar helical packing and radius profile for L and DpPorA pores, indicating identical structural pore conformation despite amino acid stereo inversion facilitating charge-selective ion and peptide transport (Supplementary Fig. 13)[40]. Moreover, most previous fluorescent imaging GUV assays were focused on antimicrobial peptides that rupture the membrane non-specifically without stable pore formation[27,44]. Here, we show the ability of DpPorA to insert and form large-diameter flexible pores in the membranes using size-dependent analysis in GUVs.

Peptides can act on cancer cells in multiple ways, and most previous anticancer peptide studies have not demonstrated the stable pore formation in the membrane[45,46]. The formation of peptide pores leading to cell death through membrane disruption is fascinating due to its ability to bypass conventional chemotherapy approaches, even if multidrug resistance mechanisms come into play[45,47]. Our work suggests that the DpPorA DE chiral peptide is more effective in forming pores in the cancer cell membrane that are highly fluidic and possess a larger surface area[48]. Notably, these highly hydrophobic and cationic peptides selectively target the negatively charged cancer cell membrane via electrostatic interactions, forming stable pores[31,49]. We have demonstrated conclusively that 5-FAM-DpPorA DE peptides enter the cancer cell membranes, which is ultimately responsible for the disruption in the membrane, leading to loss of cell viability. The use of biostable unnatural D peptides could have the added advantage of an immune tolerance induction that would be beneficial as a therapeutic strategy in vivo[31,46,48–50]. Peptides have advantages over small-molecule inhibitors and antibodies, making them attractive as anticancer molecules due to their deeper penetration properties[45,47]. Peptides can have multiple applications in targeted therapeutics, with examples ranging from delivery carriers to cell surface receptor inhibitors[45,47]. An exciting concept to explore would be combining a mirror-image pore-forming peptide with a targeted drug payload for more efficient delivery and ablation of cancer cells[47]. In the future, the molecular mechanism of the differential action of pore-forming peptides will be evaluated on diverse molecular subtypes of breast cancer cells. Importantly, the stable mirror-image pore formation across membrane models, such as planar lipid bilayers, GUVs, and mammalian cellular membranes, indicates their structural robustness, broad applicability, and membrane adaptability.

## Methods

### High-resolution single-channel recordings
High-resolution electrical recordings for the single-channel characterization were conducted in planar lipid bilayers made up of 1,2-diphytanoyl-sn-glycero-3-phosphocholine (DPhPC, Avanti Polar Lipids)[23,27,51]. The lipid bilayer is formed across a polytetrafluoroethylene (Teflon) aperture of ~100 μm in diameter[23]. Teflon is placed between the two sides of the Delrin bilayer chamber, partitioning it into cis and trans compartments (600 μL each). For constructing a solvent-free lipid bilayer, the aperture is made lipophilic by pre-painting both sides with hexadecane in n-pentane (1 μL, 5 mg mL⁻¹)[51]. After ~10 min, the chambers were filled with buffered electrolytes (1 M KCl, 10 mM HEPES, pH 7.4), and DPhPC in n-pentane (1 μL, 5 mg mL⁻¹) was added. After ~15 min incubation, the lipid bilayer is formed by raising and lowering the electrolyte solution, bringing the two lipid monolayers together at the aperture[51]. The cis compartment was connected to the ground electrode, while the trans compartment was connected to the working live electrode. Subsequently, DpPorA or DpPorA DE peptides solubilized in 0.1% DDM (1 μL, 100 μg mL⁻¹) were consistently introduced into the cis compartment of the bilayer. The peptide pore insertion was facilitated by mixing the bilayer chamber and applying a high voltage of +200 mV. The single-channel electrical properties of the peptide pores and analyte-induced pore blockages were examined in 1 M KCl, 10 mM HEPES, pH 7.4. Ion selectivity was assessed under asymmetric salt conditions with a KCl concentration gradient across the planar lipid bilayer (0.15 M cis/1 M trans) (SI text)[27]. The ion current was acquired and amplified using an Axopatch 200B amplifier, digitized with Digidata 1550B digitizer, and recorded using the pClamp 11 acquisition software (Molecular Devices, CA). The current signals were filtered with a 2 kHz low-pass filter frequency and 10 kHz sampling frequency, or with 10 kHz low-pass filter frequency and 50 kHz sampling frequency, in line with the requirement.

### Molecular dynamics simulations
The modeled pores were integrated into a 1,2-diphytanoyl-sn-glycero-3-phosphocholine (DPhPC) bilayer and solvated with TIP3P water molecules on both sides of the membrane, maintaining a water thickness of 25 Å on each side using the CHARMM-GUI Membrane Builder[52]. Subsequently, the charge of the pores was neutralized by adding K⁺ or Cl⁻ ions, followed by the addition of 1.0 M KCl to mimic experimental conditions and explore ion conductance. All systems, comprising the protein, membrane, solvent, ions, and ligand, were enclosed within a rectangular box measuring approximately $9.5 \times 9.5 \times 12$ nm³, containing roughly 100,000 atoms, whereas a larger box of $9.5 \times 9.5 \times 15$ nm³ with about 132,000 atoms was used for the pulling simulation of the nonaglutamic acid. The MD simulations were conducted using the CHARMM36-m force field within GROMACS-2021.5[53,54]. The energy minimization of the model systems was performed using the steepest descent method. Equilibration consisted of a two-step constant volume (NVT) equilibration of 5 ns each, followed by a four-step constant pressure (NPT) equilibration of 75 ns, gradually removing restraints on the protein, membrane, and ligands. The Berendsen thermostat was utilized during the NVT equilibration to control the temperature, with a temperature coupling constant of 1 picosecond (ps)[55]. This NVT equilibration allows the system to adjust to the specified temperature gradually, 300 K, while maintaining a fixed volume. Furthermore, an NPT equilibration was conducted under constant pressure conditions using the semi-isotropic coupling method coupled to a Berendsen barostat set at 1 bar, and the coupling constant for pressure was set to 2 ps. This NPT equilibration phase ensures the system achieves a stable configuration under the specified pressure conditions, allowing for accurate simulations of the subsequent dynamics. Non-bonded interactions were considered using the Verlet cut-off scheme with a cut-off of 10 Å for Coulomb and Lennard-Jones interactions, while long-range electrostatic interactions were evaluated using the Particle Mesh Ewald scheme[56]. The Linear Constraint Solver (LINCS) algorithm was employed to enforce constraints on the covalent bonds within the protein molecules[57]. Following the six-step equilibration, the systems were simulated three times for 500 ns each in an NPT ensemble using a Parrinello-Rahman barostat, semi-isotropic pressure coupling method, and Nose-Hover thermostat[58,59]. The root mean square deviations (RMSD) were computed from the trajectories relative to the equilibrated structure to evaluate the system stability comprehensively and to monitor any structural alterations occurring within the pores. The final structures resulting from the unbiased MD simulations after 500 ns were further analyzed to explore ion conductance in the presence of applied fields. Steered Molecular Dynamics (SMD) simulations were conducted by exerting a pulling force on the center of mass (COM) of the nonaglutamic molecules from the C-terminal end of the pore towards the N-terminal end along the channel axis[60]. The force profiles obtained from SMD were determined as a function of the reaction coordinate z, defined as the COM distance between the analyte and the Cα atoms of the pores. These SMD simulations were replicated three times for each analyte, maintaining a constant velocity of 0.1 Å/ns and employing a 100 kJ/mol/ns² spring constant.

## GUV-based molecular transport assays

The giant unilamellar vesicles (GUVs) were constructed using gel-assisted swelling methodology and labeled with 0.05 mol% ATTO 550 DOPE[27,44,61]. Thin lipid films were spread on polyvinyl alcohol (PVA) coated glass slides using 30 μL of a 1 mg/mL solution of DOPC (Cis and Ent lipids prepared separately). An aqueous salt solution of 0.1 M KCl and 10 mM HEPES at pH 7 was used to swell the dried lipid film, which resulted in GUV formation. The successful formation of vesicles was verified in real time by phase-contrast microscopy. To study the transport across DpPorA and DpPorA DE pores, the peptides were added to GUVs (at a concentration of 10–15 μM in 0.1% DDM) along with the fluorescent hydrophilic molecules (namely Alexa Fluor 350 Hydrazide: M.W·349 Da, Alexa Fluor 555 Hydrazide: M.W·1150 Da, ATTO 488 Dextran: M.W·3000 Da)[62]. The uptake of fluorescent molecules was monitored over 40 min, with time-lapse imaging at every minute for the first 10 min and then at every 5 min for the following 30 min, tracking either single vesicles or multiple vesicles within the same field of view. Later on, individual vesicles were further scanned and imaged to obtain datasets for statistical analysis. Control experiments were conducted under identical conditions using vesicles prepared with the same detergent concentration but without peptide incorporation. At each time point, the vesicle images were blank-subtracted, and the intensity inside the vesicle ($I_{in}$) was normalized against the intensity outside ($I_{out}$)[27]. Complete permeabilization of the vesicles was marked by $I_{in}/I_{out} \geq 0.98$ for peptide-reconstituted vesicles, while $I_{in}/I_{out} \geq 0.95$ was considered as the cut-off for permeabilization in the control vesicles.

## Cell culture and cell proliferation assay

The MDA-MB-231 cell line was obtained from ATCC (Kind gift from Dr. Annapoorni Rangarajan, Indian Institute of Science, Bengaluru, Karnataka, India) and tested negative for mycoplasma contamination. Cells were cultured in DMEM (Gibco, 11965-092) supplemented with 10% FBS (Thermo Fisher Scientific, 10270-106) and Penicillin (100 Units/mL) /Streptomycin (100 μg/mL) (Thermo Fisher Scientific, 15070-063). MCF10A cells (kind gift from Prof. Alexander Swarbrick, Garvan Institute, Sydney, Australia) were cultured in DMEM/F12 (Invitrogen, 11330-032), Horse serum (Invitrogen, 16050-122), EGF (Pepro-Tech, AF-100-15-1 mg), Hydrocortisone: (Sigma, H-0888), Cholera Toxin: (Sigma, C-8052), Insulin (Sigma, I-1882) and Penicillin (100 Units/mL) /Streptomycin (100 μg/mL) (Thermo Fisher Scientific, 15070-063). The cells are grown at 37 °C in a 5% $CO_2$ incubator. 5000 MDA-MB-231 cells or 10000 MCF10A cells were seeded into 96-well plates and were treated with appropriate concentrations of Peptide/ DDM after 24 h. Experiments were performed in biological replicates where each replicate (n) was an independently seeded and treated culture of MDA-MB-231 or MCF10A cells on different days with different passage numbers. Cell proliferation was measured 24 h after the addition of Peptide/ DDM using MTT assay (3-(4,5-dimethylthiazol-2-yl)-2,5-diphenyltetrazolium bromide, M6494, Thermo Fischer Scientific). Briefly, 10 μl MTT solution (5 mg/ml) was added to each well and incubated for 2 h at 37 °C in the dark, following which the media was removed. The purple-colored formazan crystals were dissolved by adding 100 μL DMSO (D5879, Sigma Aldrich), and the absorbance at 570 nm was measured using a plate reader (EMax plus microplate reader).

## Immunofluorescence studies

MDA-MB-231 cells were seeded in a 24-well plate (Thermo Fisher Scientific, 142475) containing sterilized coverslips (Blue star, Microscopic cover glass circular 12 mm 10 gm) at a cell density of $3 \times 10^4$ cells per well. 24 h post-seeding, cells were treated with DpPorA DE peptide, 5-FAM-DpPorA DE peptide, or a corresponding concentration of DDM. After 24 h of treatment, cells were stained with CellMask Deep Red (Thermo Fisher Scientific, C10046; 1:4000 dilution in PBS) for 5 min at 37 °C. After membrane staining, the cells were washed with PBS and

fixed in 2% PFA (Himedia, TCL119) for 5 min. After PBS wash, the coverslips were carefully removed from the wells and mounted with Pro-Long™ Gold Antifade Mountant (Invitrogen, P10144). The cells were immediately visualized within 60 min using the Carl Zeiss Axio Observer fluorescence microscope. Fluorescence was quantified by drawing the ROI around individual cells to measure the green channel intensity (5-FAM-DpPorA DE). Integrated Density was measured using ImageJ software for 50 cells per group.

## Reporting summary

Further information on research design is available in the Nature Portfolio Reporting Summary linked to this article.

## Data availability

The data that support the findings of our work are included in the main text and supplementary information files. Additional data are available from the corresponding author upon request. MD simulation data have been deposited in the Zenodo database at https://zenodo.org/records/16875053. Source data is available for Figs. 1c, i and 2c, f and Supplementary Figs. 3e, 4h and 18b in the associated source data file.

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

## Acknowledgements

This work was supported by the research grant awarded by the Department of Biotechnology, Government of India (BT/PR53613/BMS/85/242/2024) (BT/PR41427/BRB/10/1959/2020). K.R.M. thanks RGCB for the support of intramural research funding. K.R.M. thanks Dr. Krishnananda Chattopadhyay, CSIR-Indian Institute of Chemical Biology, for providing

the alpha-synuclein protein. H.B. thanks the INSPIRE Faculty Award and SERB POWER grant, Department of Science and Technology, Government of India (DST/INSPIRE/04/2020/000015 and SPG/2021/002450). H.B. acknowledges support from the following funding agencies: CSIR (FIR070301) and ICMR (IIRP-2023-0436). R.N. acknowledges intramural funding from the Centre for Human Genetics, Bangalore, India, and Ms. Jyotilakshmi S for assistance with Illustrator. Computing time was available on the high-performance computers HLRN-IV at the NHR Centers NHR@Göttingen and NHR@ZIB through the project hbp00058. These centers are jointly supported by the Federal Ministry of Education and Research and the state governments participating in the NHR. Furthermore, part of the simulations was performed on a computer cluster funded through the DFG project INST 676/7-1 FUGG.

## Author contributions

N.F. determined the biophysical properties of peptides and performed and analyzed the single-channel current recording data. K.J. performed molecular dynamics simulations supervised by the UK. R.A. and M.S. performed the cancer cell proliferation assays and cell membrane staining assays, supervised by R.N. S.R. performed and analyzed fluorescence imaging of giant vesicles, supervised by H.B. K.R.M. conceived and supervised the study. All authors wrote and approved the paper.

## Competing interests

The authors declare no competing interests.
