## [Transparent Peer Review file · Nature Communications]

Fabrication of Cytotoxic Mirror Image Nanopores

Corresponding Author: Dr Mahendran Mahendran

Version 1:

Reviewer comments:

Reviewer #1

(Remarks to the Author)

The authors present here compelling data which show that they have managed to construct mirror-image nanopores and characterize their properties, while they also assembled a larger pore with higher conductance, which can be used for a variety of biomolecules such as peptides and cyclic sugars.

Their findings are significant and will generate interest in the wider nanopore technology field and further extend its scope by paving the way for detailed structural characterisation for these of new, with unique capabilities pores (~3-4 nS) and mirrored architecture using structural (cryoEM, X-Ray) and/or spectroscopic (EPR, PELDOR/DEER, FRET) to decipher their assembly, oligomerization ensemble (i.e. the "unstable hexameric pores") and their conformational dynamics during molecular transport.

Particular points to address:

- The pore diameters presented in the atomistic MD simulations of ~ 10-15 Å for Lp/DpPorA and Lp/DpPorA DE would be expected to result in single channel conductances of ~1 nS, consistent with helical (rod shaped) membrane pores. The reported conductances though here are much higher ~3 nS and 4 nS respectively. This indicates that the pores in the electrophysiology recording set up adopt a much larger opening diameter (i.e. > 25 Å) as reported for the large conductance mechanosensitive channel MscL. Can the authors explain this discrepancy between single channel recordings and MD data. Can they suggest or implement approaches that this could be resolved so visualisation of these pore states is possible to appreciate their full extent and capabilities.
- From the MD radius graphs it seems that the channel "gate" shifts along the Z-axis (away from "0") between the two types of pores in opposite directions. Can the authors explain why is that and if this would have an effect on molecular transport.
- ~ 200 pN pulling forces reported by MD. How biologically relevant these values are. Has something similar previously been measured experimentally.
- "DDM concentrations up to 0.007% did not have a significant cytotoxic effect", which is somewhat expected given that DDM's CMC is 0.0089%, therefore the non-cytotoxic DDM concentrations of 0.007% are below the CMC, which means that DDM cannot form micelles and possibly keep the peptides into solution (for which DDM is essential), so they do not precipitate and form these pores. Have the authors tested peptide solubility and folded pore assemblies by SEC or any other methods to conclude the optimal in cell pore incorporation below the DDM CMC concentration (i.e. < 0.0089 %). This will ensure that peptides are properly solubilised and not aggregated, immediately prior they are added to the cells. I reckon, after peptides are added to the cells the DDM concentration drops dramatically and well below the CMC, but this should not be a problem anymore given the cell membrane lipids will now rescue (by incorporation) the majority? of those peptides and prevent aggregation
- Is it possible to quantify the % level of peptide reconstitution into the cell membranes, i.e. how much peptide is initially added (solubilised in DDM buffer) and how much is eventually incorporated into the cell membranes. A % of the added peptide may aggregate, or unspecifically bind to cells or may not be incorporated at all. I believe some % values here would hold to evaluate and explore the cytotoxicity effect for biomedical applications.

Minor points to address:

Abstract: define what "real" pores mean, as this is the main novelty offered by this study here

Reviewer #2

(Remarks to the Author)

"Fabrication of Cytotoxic Mirror Image Nanopores" by N. F. CA et al. is an article that characterizes the single molecule sensing capabilities of mirror-image nanopores DpPorA and DpPorA DE nanopores. Molecular dynamic simulations analyze the pulling force of these nanopores on E9 and PEG-E9 translocations. Finally, they demonstrate the effect of these peptides on cancer cells and show the mirror image peptides can reduce the cell viability by nearly 20%. Overall, my assessment of this work is that it is a well written paper and a thorough study of the mirror image peptides. The combination of single molecule detection with the pores (Fig. 2g, h, i), the MD simulation results (Figs. 3, 4), the membrane viability studies (Fig. 5) and the cancer cell demonstration (Fig. 6) represent a thorough analysis of the aforementioned peptides. I believe the paper should be published, but I'm not convinced the novelty of these results warrants publication in Nature Communications. It seems more appropriate for a specialized journal or perhaps Scientific Reports.

Some comments for further consideration before publication:

1. There is a significant amount of "gating" with the "DE" nanopores in Fig. 2. It would be nice to see what the current fluctuations look like prior to addition of the various analytes in Fig. 2g (E9), 2h (PEG200-E9) and 2i (alpha synuclein).

2. Extended Fig. 7b and 7c, Supp. Fig. 9: Ideally, one would like to see a significant decrease in the cell viability with the addition of the DpPorA DE peptides and this is reported in Fig. 6, but the authors report the addition of only 10 and 20 uM concentrations in extended fig. 7 and Supp. Fig. 9. I see some decrease in viability here, but not much. This leads me to a question and a comment.

Question: Why did the authors stop at 20 uM? It seems like it would be better to further increase the concentration of the peptides until you see a much more significant decrease in cell viability (like the 25 uM concentration in Fig. 6). There is a comparison to the DDM concentration, but DDM is a detergent and thus maybe not a useful comparison to the peptides. For Supp. Fig. 9, wouldn't it be better to compare the impact that the D-peptides and L-peptides have on the cancer cells to show that the mirror image peptides are far more effective at killing the cancer cells. I don't quite follow the rationale for comparing the peptides to the DDM detergent.

Comment: For Supp. Fig. 9, panel A states there is a $p < 0.01$ overlap between the DDM and DpPorA DE additions while panel C states there is no significant difference. A cursory view of the figure leads me to believe that is probably true although there seems to be a sizable amount of overlap in the 20 uM bars in panel A, so I guess I have some difficulty agreeing that this is indeed a $p < 0.01$ value. In any event, I think it would be best to clearly state the p-value for the panel C overlap rather than just stating it is not significant.

Version 2:

Reviewer comments:

Reviewer #1

(Remarks to the Author)

I am satisfied with the additional experiments, calculations and the rewriting undertaken to address most of my concerns and suggestions.

Minor point: Include the full SEC profile in Extended Figure 8. This is essential to conclude for any occurring aggregation at earlier elution volumes (should appear ~8-9ml depending which column that is). Further, state which particular column was used in this case (Superdex 200 10/300?) and calculate the expected sizes (in kDa) of your peaks, including the peak appearing at ~14ml (is this a lower-oligomeric/monomeric peptide state?). An SDS Gel of the individual SEC peaks (i.e. 2x @12ml and 1x @14 ml) would be informative here.

Reviewer #2

(Remarks to the Author)

The authors have done an excellent job addressing my comments and concerns. I am now satisfied that this work is sufficiently novel for publication in Nature Communications. I am also satisfied with the manner in which they addressed my additional comments. This is very nice work. Congratulations!

We would like to thank all reviewers for their comments and suggestions. We have addressed each comment and made changes that have significantly improved the revised manuscript.

Reviewer 1:

The authors present here compelling data which show that they have managed to construct mirror-image nanopores and characterize their properties, while they also assembled a larger pore with higher conductance, which can be used for a variety of biomolecules such as peptides and cyclic sugars.

Their findings are significant and will generate interest in the wider nanopore technology field and further extend its scope by paving the way for detailed structural characterisation for these of new, with unique capabilities pores (~3-4 nS) and mirrored architecture using structural (cryoEM, X-Ray) and/or spectroscopic (EPR, PELDOR/DEER, FRET) to decipher their assembly, oligomerization ensemble (i.e. the “unstable hexameric pores”) and their conformational dynamics during molecular transport.

Particular points to address:

1. The pore diameters presented in the atomistic MD simulations of ~ 10-15 Å for Lp/DpPorA and Lp/DpPorA DE would be expected to result in single channel conductances of ~1nS, consistent with helical (rod shaped) membrane pores. The reported conductances though here are much higher ~3 nS and 4 nS respectively. This indicates that the pores in the electrophysiology recording set up adopt a much larger opening diameter (i.e. > 25 Å) as reported for the large conductance mechanosensitive channel MscL. Can the authors explain this discrepancy between single channel recordings and MD data. Can they suggest or implement approaches that this could be resolved so visualisation of these pore states is possible to appreciate their full extent and capabilities.

We thank the reviewer for this insightful observation. The conductivity of a double-cone cylindrical pore can be approximately calculated using the following formula:

$$G = \frac{k\pi dD}{4L}$$

where k is the specific conductivity of the electrolyte KCl, 1.1 nS/Å, d is the diameter of the constriction zone, D is the wider diameter, near the terminal of the pore, and L is the length of the pore. If we use $d = 10 \text{ Å}$, $D = 30 \text{ Å}$, and $L = 80 \text{ Å}$, the calculated conductivity is approximately 3.25 nS. Depending on the parameters, this value can vary, but it shows

that the conductance values obtained by the MD simulations are indeed reasonable (Siwy, Z.S. Adv. Funct. Mater. 2006; Apel, Pavel Yu, et al. Nanotechnology 2012).

Figure R1: Schematic representation of the double-cone cylindrical pore. d is the diameter of the constriction zone, D is the wider diameter, near the terminal of the pore, and L is the length of the pore.

As suggested, we have now included a new figure that clearly shows the full extent of the pore geometry along the channel axis. The HOLE radius profile was reproduced with the surface for better clarity of how the designed pores form the double-cone structures. The constriction zone is primarily shaped by the spatial distribution of bulky amino acid residues lining the inner pore wall, with LYS24 playing a central role in defining this region. In LpPorA and DpPorA, the basic LYS24 residues form salt bridges with the acidic ASP28 residues, resulting in an angled orientation of these side chains relative to the Z-axis. In contrast, in LpPorA DE and DpPorA DE, where the salt-bridge interactions are missing due to the mutation of ASP28 with alanine, the LYS24 side chains adopt a more perpendicular alignment to the Z-axis. This perpendicular, inward-facing arrangement results in a narrower constriction zone for LpPorA DE and DpPorA DE. Nevertheless, these variants exhibit wider openings at the vestibular regions, which contributes to their slightly higher single-channel conductance compared to LpPorA and DpPorA. It is worth mentioning that the surface radius profile was only generated from the 500 ns final structure of all the pores, whereas figures 3c and 3f were obtained by averaging over 100 structures. We have discussed this in the revised manuscript in detail (Main text pages 14, 15, and new Supplementary Fig. 8).

Figure R2: The HOLE surface-radius profile along with the designed pore.

a. The surface HOLE profile is generated from the 500 ns final structure of the LpPorA pore. The angled orientation of the salt bridge interacting residues LYS24 and ASP28 are depicted with “CPK model” where the pore is shown with a tube. **b.** The surface hole profile of DpPorA. **c.** The surface hole profile of LpPorA DE. The absence of the salt bridge interactions allowed the LYS24 residues to be oriented perpendicular to that of the Z-axis. **d.** A similar HOLE radius profile was observed for the DpPorA DE pore.

- From the MD radius graphs it seems that the channel “gate” shifts along the Z-axis (away from “0”) between the two types of pores in opposite directions. Can the authors explain why is that and if this would have an effect on molecular transport.

Indeed, our MD-derived pore radius profiles show that the position of the channel constriction (the “gate”) shifts along the Z-axis in opposite directions for the two pore designs. This shift arises due to asymmetries in the molecular architecture of the pores, specifically, the removal of acidic amino acid residues in the mutant D/LpPorA DE shifted

the gate to the opposite direction to that of the wild-type D/LpPorA. In the wild-type D/LpPorA, steric or electrostatic constraints lead to a central narrowing toward positive Z, while the mutant D/LpPorA DE pores favor a shift toward negative Z. The positional asymmetry influences the molecular transport by altering the local potential energy landscape and hydration environment, and is also reflected in the slightly different conductivity values of these pores. However, in our simulations, we observe that although the gate location varies, the effective translocation time and peptide alignment do not differ significantly between the two pores under similar applied-field conditions. We have now added a clarification in the manuscript discussing this shift and its potential functional consequences (Main text pages 14, 15 and new Supplementary Fig. 8).

3. ~ 200 pN pulling forces reported by MD. How biologically relevant these values are. Has something similar previously been measured experimentally.

First of all, we have to mention that we are pulling with a velocity of 0.1 Å/ns which is slow in terms of simulations but likely fast in terms of experiment. Although the pulling speed might lead to forces that are too high in absolute numbers, the force profile as a function of the channel axis contains information on the translocation. Notably, many experimental studies using atomic force microscopy (AFM) have reported that unfolding individual protein domains, such as titin immunoglobulin domains, typically requires forces in the range of 150–300 pN, depending on the pulling rate and protein structure. Thus, the ~200 pN pulling forces observed in our MD simulations fall within the biologically realistic regime for mechanical manipulation of protein or peptide backbones. These values are consistent with unfolding transitions involving β -sheet structures or tightly packed α -helices under relatively fast pulling speeds. While AFM studies of protein unfolding are well established, there is an absence of direct experiments on the forces required to translocate peptides through nanopores. This gap is due to the technical challenges associated with such experiments. Specifically, precise alignment of the pulling vector through the narrow and curved axis of a nanopore is extremely difficult. Calibration of AFM cantilevers at the low force sensitivity (1–20 pN) needed to resolve the translocation forces is prone to uncertainty. Dynamic interactions between the peptide and the pore, including friction, electrostatics, and conformational fluctuations, make it hard to isolate a clean signature. As a result, steered molecular dynamics (SMD) simulations have become the primary tool to investigate such forces at atomic resolution. (Rief, M., et al.

Science 1997, Merkel, R., et al. Nature 1999, and Scheuring, S. Proc. Natl. Acad. Sci. 2023). We have discussed this in the revised manuscript (Main text page 16).

4. "DDM concentrations up to 0.007% did not have a significant cytotoxic effect", which is somewhat expected given that DDM's CMC is 0.0089%, therefore the non-cytotoxic DDM concentrations of 0.007% are below the CMC, which means that DDM cannot form micelles and possibly keep the peptides into solution (for which DDM is essential), so they do not precipitate and form these pores. Have the authors tested peptide solubility and folded pore assemblies by SEC or any other methods to conclude the optimal in cell pore incorporation below the DDM CMC concentration (i.e. $< 0.0089\%$). This will ensure that peptides are properly solubilised and not aggregated, immediately prior they are added to the cells. I reckon, after peptides are added to the cells the DDM concentration drops dramatically and well below the CMC, but this should not be a problem anymore given the cell membrane lipids will now rescue (by incorporation) the majority? of those peptides and prevent aggregation.

Thank you. This is a brilliant observation from the reviewer.

- ✓ We examined the pore-forming activity of DpPorA DE peptides solubilized in low concentrations of DDM (0.007% and 0.0089%). The peptides consistently formed pores that exhibited similar single-channel electrical properties in all tested conditions, indicating structural stability. Notably, peptides solubilized in 0.1% DDM inserted rapidly into the membrane, and this condition was therefore selected for all single-channel experiments, as demonstrated in the main figure 1 and figure 2. We have included this data and discussed it in the revised manuscript (Main text page 25 and new Supplementary Fig. 4).

Figure R3: Single-channel properties of DpPorA DE in different DDM concentrations. Single-channel electrical recording of DpPor DE with different DDM concentrations in 1 M KCl buffer, 10 mM HEPES, pH 7.4. The current signals were digitally filtered at 2 kHz.

✓ Further, as suggested by the reviewer, we assessed the solubility and folding of DpPorA peptides by analytical size exclusion chromatography (SEC) using different DDM concentrations (0.008% and 0.1%). Peptides solubilized in 0.1% DDM produced a sharp peak in SEC corresponding to peptide folding, consistent with stable pore formation in single-channel recordings. To optimize for biological compatibility in cellular assays, we tested peptide solubilization in 0.008% DDM, which is closer to the critical micelle concentration. Under these conditions, SEC still showed a small peak, indicating that a portion of the peptides remained folded even at lower DDM concentration, in agreement with single-channel recording data. We believe this folding level is sufficient for initiating functional studies, particularly anticancer activity assays in cells. We have included a new figure showing that peptide aggregation was not observed till 25 μ M concentration was used in these experiments (Main text page 19, Extended Data Fig. 8 and Supplementary Table 2).

Figure R4: Effect of DDM on pPorA peptides and MDA-MB-231 cells.

*a. Analytical SEC of DpPorA DE in 0.1% DDM and 0.008% DDM. b. MDA-MB-231 cells were treated with DDM concentrations ranging from 0.001% to 0.1% for 24 hours, and cell viability was measured using the MTT assay. Viability decreased with increasing DDM concentrations. The viability was comparable to control till 0.007% DDM, while concentrations above 0.007% showed significant cytotoxic effects. Concentrations below 0.007% are non-detrimental to cell viability. $n=3$, $**p<0.01$, $***p<0.0001$. c. Microscopy images showing that peptide aggregation was not observed till 25 μM concentration used in the experiments (20X magnification) Scale:20 μm .*

- ✓ Importantly, we agree with the reviewer's brilliant observation that after peptide addition to cell cultures, the DDM concentration rapidly falls below its CMC due to dilution. Furthermore, under these low-detergent conditions, we expect that the peptides will be stabilized by incorporation into the lipid bilayer of the cell membrane, which can serve as a native-like environment to support peptide folding and prevent aggregation. We have included this new data and discussed it in the revised manuscript (Main text page 19, Extended Data Fig. 8 and Supplementary Table 2).
5. Is it possible to quantify the % level of peptide reconstitution into the cell membranes, i.e. how much peptide is initially added (solubilised in DDM buffer) and how much is eventually incorporated into the cell membranes. A % of the added peptide may aggregate, or unspecifically bind to cells or may not be incorporated at all. I believe some % values here would held to evaluate and explore the cytotoxicity effect for biomedical applications.

Thank you for this excellent suggestion. To address this query, we have synthesized and characterized the functional activity of FAM-labeled peptides through single-channel electrical recordings and cancer cell assays.

- ✓ We investigated the pore-forming ability of the DpPorA DE peptide tagged with 5-carboxyfluorescein (5-FAM) using single-channel recordings. The fluorescently labeled peptide rapidly inserted into the bilayers and formed well-defined, stable pores at different voltages. The single-channel properties of the pore were identical to the untagged DpPorA DE peptide. This indicates that tagging the peptide with 5-FAM does not alter its pore-forming mechanism. Based on this data, the fluorescently labeled DpPorA DE peptides were subsequently used in cellular studies to evaluate their anticancer activity (Main text page 22, Figure 6 and Extended Data Fig. 9).

Figure R5: Single-channel properties of 5-FAM-DpPorA DE.

a. Single-channel insertion of 5-FAM-DpPorA DE at +200 mV, with corresponding current-amplitude histogram as inset. **b.** Electrical recording showing characteristic gating of 5-FAM-DpPorA DE at +100 mV. **c.** I-V curve of 5-FAM-DpPorA DE showing stable current from -100 mV to +100 mV. Electrical recording showing stable current trace of 5-FAM-DpPorA DE at **d.** +20 mV, **e.** +30 mV, and **f.** +50 mV, with corresponding current-amplitude histogram as inset. The current signals were digitally filtered at 2 kHz. Electrolyte: 1 M KCl, 10 mM HEPES, pH 7.4.

- ✓ We next quantified the incorporation of 5-FAM-DpPorA peptides into the living cancer cell membrane (MDA-MB-231 cells). These peptides incorporated into the cell membrane within 4 hours, as visualized by the green channels in the cell membrane, with increasing intensity observed by 24 hours. For example, a single-cell fluorescence intensity scatter plot confirms time-dependent peptide incorporation in the cell population. We included this new data and discussed it in the revised manuscript (Main text page 2, 21, 22, 24, main Figure 6e, 6f and Extended Data Fig. 10).

Figure R6 5-FAM-DpPorA DE peptide action on MDA-MB-231 cells

a. Representative fluorescence images showing the distribution of 5-FAM-DpPorA DE (green) in MDA-MB-231 cells at 4 hours and 24 hours post-addition of peptide. Cell membranes were stained with CellMask (red) (scale bar 10 μm , magnification 63X). **b.** Single-cell fluorescence intensity scatter plot showing increased incorporation of 5-FAM-DpPorA DE at 24 hours compared to 4 hours, indicating time-dependent peptide incorporation in the cell population. **c.** Representative fluorescence images showing the distribution of 5-FAM-DpPorA DE (green) in MDA-MB-231 cells at 4 hours and 24 hours post-treatment. Scale bar: 20 μm . Cell membranes were stained with a CellMask dye (red). Merged images highlight the co-localization of 5-FAM-DpPorA DE with the plasma membrane. Scale bar: 10 μm . **d.** Quantification of 5-FAM-DpPorA DE fluorescence intensity at 4 hours and 24 hours using ImageJ, representing the average fluorescent signal at different time points.

- Abstract: define what “real” pores mean, as this is the main novelty offered by this study here

Thank you for your suggestion. Our previous studies demonstrated pore formation using mirror-image peptides; however, distinct single-channel properties between L and D pores (Krishnan et al., Nat. Commun., 2024) indicated they were not mirror-image structures. We attributed this to a cysteine residue at the 24th position, promoting pre-oligomerization of

peptide monomers into octamers before membrane insertion. In contrast, the DpPorA and DpPorA DE peptides used in this study lack cysteine residues, forming pores only upon membrane insertion. These D-peptides exhibited pore conductance and selectivity nearly identical to their L counterparts, confirming the formation of mirror-image pores. MD simulations further supported this, revealing highly similar helical packing and pore radius profiles for L and DpPorA. In the revised manuscript, we now describe these as fully functional mirror-image pores to better emphasize their novelty (Main text page 2 and 3).

Reviewer 2 comments:

Fabrication of Cytotoxic Mirror Image Nanopores” by N. F. CA et al. is an article that characterizes the single molecule sensing capabilities of mirror-image nanopores DpPorA and DpPorA DE nanopores. Molecular dynamic simulations analyze the pulling force of these nanopores on E9 and PEG-E9 translocations. Finally, they demonstrate the effect of these peptides on cancer cells and show the mirror image peptides can reduce the cell viability by nearly 20%. Overall, my assessment of this work is that it is a well written paper and a thorough study of the mirror image peptides. The combination of single molecule detection with the pores (Fig. 2g, h, i), the MD simulation results (Figs. 3, 4), the membrane viability studies (Fig. 5) and the cancer cell demonstration (Fig. 6) represent a thorough analysis of the aforementioned peptides. I believe the paper should be published, but I’m not convinced the novelty of these results warrants publication in Nature Communications. It seems more appropriate for a specialized journal or perhaps Scientific Reports.

We thank the reviewer for their comments and would like to first clarify the novelty of our work before addressing the specific points raised.

- ✓ Our study represents a significant conceptual and technical advancement in constructing synthetic alpha-helical pores at both the molecular and cellular levels. Using a multidisciplinary approach combining electrical recordings, mutational analysis, molecular dynamics simulations, fluorescence assays, and cancer cell assays, we confirm the formation of a functional mirror-image nanopore for sensing and biomedical applications. We have highlighted key points below demonstrating the novelty of our work.
- ✓ Computational design has enabled the de novo creation of α -helical pores, as demonstrated by pioneering work from the Baker group (Xu et al., Nature, 2020) and our collaborative efforts with the Woolfson group (Scott et al., Nature Chemistry, 2021; Niitsu et al., JACS,

2025). However, these pores are limited by low conductance and non-uniform channels. Furthermore, in our previous work (Krishnan et al. Nat. Commun., 2022), we constructed alpha-helical pores using mirror-image peptides that exhibited distinct conductance states (1.5 and 4.0 nS), indicating heterogenous assemblies, likely limiting their nanopore application (Main text page 3).

- ✓ Our study represents the first report on true mirror-image pores with identical conductance profiles by L and D pores, unlike those reported earlier by our group (Krishnan et al. Nature Communications, 2022). We further confirmed the design of uniform mirror-image pores by demonstrating the identical properties of L and D-DE mutant pores supported by MD simulations. Moreover, we rationally engineered peptides to increase conductance (~4.0 nS) and selectivity, enhancing their nanopore potential. This tuning of pore dimensions and interactions at the molecular level leads to sophisticated functional architectures.

We have now included new data showing stable channel activity under varying salt conditions, highlighting the pore's structural robustness and potential for sensing and cell-based assays (Main text pages 2, 3, 11 and Supplementary Fig. 4).

- ✓ We report the first synthetic nanopore capable of detecting small peptides, PEGylated peptides and disordered proteins. Specifically, we report the label-free, single-molecule detection of full-length alpha-synuclein (α S), a 140-residue protein central to Parkinson's disease, using mirror-image pores. Our approach captured the full α S in native conformation compared to prior strategies where only truncated and chemically labelled α S peptides were detected (Cao et al, ACS Nano, 2023; Liu et al, ACS Nano, 2023).

We have added new data showing competitive binding of E9 peptides, PEGylated peptides, and α S with the mirror-image pore. Each analyte displayed a mutually exclusive current signal, indicating selective recognition. This confirms structural compatibility and functional advantage of our pores over biological and other synthetic pores for advanced nanopore sensing (Main text page 2, 7-10, 11 and Figure 2i).

Figure R7: Competitive analyte binding with DpPorA DE pore.

a. Interaction of 100 μM E9 with DpPorA DE on the cis side at +40 mV. **b.** Interaction of 500 μM PEG 200-E9 with DpPorA DE on the cis side at +40 mV. **c.** Interaction of 25 nM alpha-synuclein with DpPorA DE on the cis side at +40 mV. Insets show corresponding dwell time (τ_{off}) histogram fitted with a monoexponential probability function, current amplitude histogram and recording at an expanded time scale. **d.** Competitive interaction of E9, PEG 200-E9 and alpha-synuclein with DpPorA DE on the cis side at +40 mV. The current signals were filtered at 10 kHz and sampled at 50 kHz. Electrolyte: 1 M KCl, 10 mM HEPES, pH 7.4.

- ✓ For transport studies in giant unilamellar vesicles, we have, for the first time, assembled peptides within enantiomeric lipid membranes to investigate their functional compatibility, which is challenging in other membrane mimetic platforms. This unique platform enables investigation of mirror-image peptides interactions in chiral lipid environments, opening new avenues for studying chirality-dependent membrane assembly and transport. We have modified the Extended Data Fig. 7 to highlight this (Main text page 19).
- ✓ This work advances the field of pore-forming peptides by introducing chemically synthesized mirror-image (D-form) α -helical DpPorA peptides that are capable of forming

functional transmembrane pores on cancer cells, unlike earlier studies. **We have now included new data to demonstrate conclusively that these 5-FAM-DpPorA DE peptides enter the cancer cell membranes, which is ultimately responsible for the disruption in the membrane,** leading to loss of cell viability. Our unique model system provides a foundation for elucidating the molecular mechanisms of pore-forming peptides and enables future studies, including combination therapies with existing anti-cancer drugs (Main text page 22, 24, main Figure 6 and Extended Data Fig. 10).

- ✓ The demonstration of consistent pore formation and function across membrane models, including **planar lipid bilayers, GUVs, and cellular membranes,** indicates the robustness of the mirror-image peptide pores and confirms the broad applicability and membrane adaptability of designed peptides (Main text page 24, 25).

Some comments for further consideration before publication:

1. There is a significant amount of “gating” with the “DE” nanopores in Fig. 2. It would be nice to see what the current fluctuations look like prior to addition of the various analytes in Fig. 2g (E9), 2h (PEG200-E9) and 2i (alpha synuclein).

Voltage-dependent gating is an intrinsic property observed in most nanopores, with the threshold potential for gating varying between different channels. However, the precise gating mechanism remains unknown. In this study, the pPorA pores exhibited voltage-dependent gating only at higher voltages (above ± 100 mV), where the channel transitioned to a fully closed state. At lower voltages, up to ± 50 mV, the channels remained fully open and did not exhibit gating behavior. To ensure accurate analyte detection without interference from gating events, we conducted all analyte blockage experiments at voltages less than ± 50 mV, where no gating was observed. In response to the reviewer’s suggestion, we have now included a new figure showing representative ion current traces prior to analyte addition and with analyte addition in the revised manuscript and also clarified the gating of the nanopores. (Please see main text page 9, Extended Data Fig. 2 and new Extended Data Fig. 3)

Figure R8: *Electrical recordings of DpPorA DE without analytes and with analytes.*

a. *Electrical recording of DpPorA DE without and with 100 μM E9 added to the cis side of pore at +40 mV. b. Electrical recording of DpPorA DE without and with 500 μM PEG 200-E9 added to the cis side of pore at +40 mV. c. Electrical recording of DpPorA DE without and with 25 nM alpha synuclein added to the cis side of pore at +40 mV. The current signals were filtered at 10 kHz and sampled at 50 kHz. Electrolyte: 1 M KCl, 10 mM HEPES, pH 7.4.*

- Extended Fig. 7b and 7c, Supp. Fig. 9: Ideally, one would like to see a significant decrease in the cell viability with the addition of the DpPorA DE peptides and this is reported in Fig. 6, but the authors report the addition of only 10 and 20 μM concentrations in extended fig. 7 and Supp. Fig. 9. I see some decrease in viability here, but not much. This leads me to a question and a comment.

Question: Why did the authors stop at 20 μM ? It seems like it would be better to further increase the concentration of the peptides until you see a much more significant decrease in cell viability (like the 25 μM concentration in Fig. 6). There is a comparison to the DDM concentration, but DDM is a detergent and thus maybe not a useful comparison to the peptides. For Supp. Fig. 9, wouldn't it be better to compare the impact that the D-peptides and L-peptides have on the cancer cells to show that the mirror image peptides are far more

effective at killing the cancer cells. I don't quite follow the rationale for comparing the peptides to the DDM detergent.

- ✓ We did not see a significant change in the cell viability upon addition of wild-type DpPorA and LpPorA peptides, and hence did not proceed with them. Notably, these peptides are neutral in charge, which most likely limits their interaction with the negatively charged cancer cell membranes. Interestingly, DpPorA DE chiral peptide (25 μM) is more effective in forming pores in the cancer cell membrane that are highly fluidic and possess a larger surface area. Notably, these highly hydrophobic and cationic peptides selectively target the negatively charged cancer cell membrane via electrostatic interactions, forming stable pores. However, we did not use higher concentrations of these peptides as we observed aggregate formation, which interfered with downstream assays by affecting solution homogeneity (see Figure R9, panel B). We have discussed this in the revised manuscript (Main text page 20, 24, main text Figure 6, Extended Data Fig. 8, Supplementary Fig. 11, and Supplementary Table 3).

Figure R9: *a.* % viability of MDA-MB-231 cells was determined using the MTT assay 24 hours after the addition of DpPorA and LpPorA peptide at 10 μM and 20 μM concentrations. $n=3$. *b.* Formation of deposits with higher concentrations of peptide, which interfered with downstream assays on MDA-MB-231 cells. Scale bar: 20 μm .

- ✓ We thank the reviewer for their suggestion to compare the impact that the D-peptides and L-peptides on the cancer cells. We performed a comparative analysis of the impact of DpPorA DE and LpPorA DE peptides on cancer cell viability. Our results clearly demonstrate that the DpPorA DE peptides exhibit a significantly greater anticancer effect compared to their LpPorA DE peptides. To illustrate this enhanced potency, we have updated the relevant data and included the revised plots to highlight the high stability and bioactivity of DpPorA DE peptides, further supporting their potential as therapeutic agents. (Main text page 20 and Supplementary Fig. 11).
- ✓ DDM facilitates peptide solubilization and folding, which in turn promotes stable pore formation. We have provided a detailed explanation of the effects of DDM on peptide pore formation in planar lipid bilayers and cellular membranes in response to Reviewer 1, Comment 4. We kindly ask the reviewer to refer to this detailed response (see main text page 19, Extended Data Fig. 8, and Supplementary Table 2).

Figure R10: The % viability of MDA-MB-231 cells was determined using the MTT assay 24 hours after the addition of LpPorA DE and DpPorA DE peptides at 10 µM, 20 µM and 25 µM concentrations. $n=3$, $***p=0.0003$. Graph demonstrating a significant decrease in viability of cells treated with the DpPorA DE compared to LpPorA DE peptide in MDA-MB-231 cells.

Comment: For Supp. Fig. 9, panel A states there is a $**p<0.01$ overlap between the DDM and DpPorA DE additions while panel C states there is no significant difference. A cursory

view of the figure leads me to believe that is probably true although there seems to be a sizable amount of overlap in the 20 uM bars in panel A, so I guess I have some difficulty agreeing that this is indeed a $p < 0.01$ value. In any event, I think it would be best to clearly state the p-value for the panel C overlap rather than just stating it is not significant.

Thank you. We have updated panel B of Supplementary Fig. 11 to clearly state the p-value (previously panel C of Supplementary Fig. 9). It shows the effect of DpPorA DE peptide on normal mammary gland cells as represented by MCF10A cells. There is no significant effect of the peptide on the normal cells, which is beneficial in a therapeutic context. (Main text page 2, 20 and Supplementary Fig. 11)

Figure R11: Cell viability of DpPorA DE on MCF10A. There was no significant effect of 20 μ M DpPorA DE on MCF10A cells as determined by MTT assay. $n=3$, ns- non-significant.

Reviewer 1: (Remarks to the Author):

I am satisfied with the additional experiments, calculations and the rewriting undertaken to address most of my concerns and suggestions.

Thank you.

Minor point: Include the full SEC profile in Extended Figure 8. This is essential to conclude for any occurring aggregation at earlier elution volumes (should appear ~8-9ml, depending on which column that is). Further, state which particular column was used in this case (Superdex 200 10/300?) and calculate the expected sizes (in kDa) of your peaks, including the peak appearing at ~14ml (is this a lower-oligomeric/monomeric peptide state?). An SDS Gel of the individual SEC peaks (i.e. 2x @12ml and 1x @14 ml) would be informative here.

The analytical SEC was primarily employed to assess peptide folding and stability. The full SEC profile is now provided, showing no detectable aggregation at the void volume (~8 mL and earlier). All SEC experiments were performed using a Superdex 200 Increase 10/300 GL column. To characterize the observed peaks, SEC fractions collected between 11 mL and 16 mL were analyzed by SDS-PAGE. These fractions showed monomeric bands at ~4 kDa, consistent with the expected peptide size. The corresponding gel images are now included alongside the SEC profile (previously Extended Figure 8, now Supplementary Figure 18).

Reviewer #2 (Remarks to the Author):

The authors have done an excellent job addressing my comments and concerns. I am now satisfied that this work is sufficiently novel for publication in Nature Communications. I am also satisfied with the manner in which they addressed my additional comments. This is very nice work. Congratulations!

Thank you.